# Methyl transfer in psilocybin biosynthesis

Jesse Hudspeth [1,2,7], Kai Rogge [3,4,7], Sebastian Dörner[3,4], Maximilian Müll[5], Dirk Hoffmeister [3,4], Bernhard Rupp [1,6] & Sebastiaan Werten [1] ✉

Psilocybin, the natural hallucinogen produced by *Psilocybe* ("magic") mushrooms, holds great promise for the treatment of depression and several other mental health conditions. The final step in the psilocybin biosynthetic pathway, dimethylation of the tryptophan-derived intermediate norbaeocystin, is catalysed by PsiM. Here we present atomic resolution (0.9 Å) crystal structures of PsiM trapped at various stages of its reaction cycle, providing detailed insight into the SAM-dependent methylation mechanism. Structural and phylogenetic analyses suggest that PsiM derives from epitranscriptomic N⁶-methyladenosine writers of the METTL16 family, which is further supported by the observation that bound substrates physicochemically mimic RNA. Inherent limitations of the ancestral monomethyltransferase scaffold hamper the efficiency of psilocybin assembly and leave PsiM incapable of catalysing trimethylation to aeruginascin. The results of our study will support bioengineering efforts aiming to create novel variants of psilocybin with improved therapeutic properties.

Psilocybin (Fig. 1) is the principal natural product of a polyphyletic group of fungi colloquially referred to as magic mushrooms[1–3]. The phosphorylated indolethylamine is a chemically stable precursor of the psychotropic substance psilocin, which acts as a partial agonist of human 5-hydroxytryptamine (5-HT) receptors[4–6]. Therapeutic evaluation of psilocybin has revealed a remarkable capacity to alleviate a variety of psychological conditions, including major depressive disorder[7], substance dependence[8] and end-of-life anxiety[9]. As a result, psilocybin has received breakthrough therapy status by the US Food and Drug Administration and is expected to enter phase III clinical trials shortly.

The rising interest in potential medical applications has prompted efforts to produce psilocybin biotechnologically, as well as to explore the properties of novel analogues. Four enzyme-encoding genes required for the biosynthetic pathway from ʟ-tryptophan to psilocybin (*psiD*, *psiH*, *psiK* and *psiM*) were recently identified in *Psilocybe cubensis* (*P. cubensis*) and other species[10–13], paving the way for large-scale heterologous production in microorganisms. The feasibility of this approach was first demonstrated in *Aspergillus nidulans* (*A. nidulans*)[14]

and subsequently also in *Escherichia coli* (*E. coli*)[15] and *Saccharomyces cerevisiae*[16]. Furthermore, in vitro biocatalytic synthesis of psilocybin from 4-hydroxyindole and ʟ-serine was reported[17], as was the biosynthetic production of ring-methylated analogues[18].

The final reaction in the psilocybin pathway is catalysed by PsiM, an *S*-adenosyl-ʟ-methionine (SAM) dependent methyltransferase (Fig. 1). This enzyme dimethylates the primary amine of the precursor norbaeocystin, via the monomethylated intermediate baeocystin, to produce the tertiary amine psilocybin[10]. In addition, PsiM-mediated synthesis of the trimethylated quaternary ammonium analogue, aeruginascin, has been postulated[16,19]. The latter compound was detected in several psilocybin-producing fungi[20–22] as well as in transgenic yeast expressing the four psilocybin biosynthesis genes[16].

In this work, we present a comprehensive X-ray crystallographic, biochemical and phylogenetic characterisation of of PsiM. Crystal structures obtained in the presence of various cofactor analogues, substrates and products provide atomic-resolution insight into the reaction cycle and explain the specificity of the enzyme. We also show that PsiM does not catalyse a third methylation reaction to produce

[1]Institute of Genetic Epidemiology, Medical University of Innsbruck, Innsbruck, Austria. [2]Department of Chemistry, Colorado School of Mines, Golden, CO, USA. [3]Institute of Pharmacy, Friedrich Schiller University, Jena, Germany. [4]Research Group Pharmaceutical Microbiology, Leibniz Institute of Natural Product Research and Infection Biology, Hans Knöll Institute, Jena, Germany. [5]Research Group Biosynthetic Design of Natural Products, Leibniz Institute of Natural Product Research and Infection Biology, Hans Knöll Institute, Jena, Germany. [6]k.-k. Hofkristallamt, San Diego, California, USA. [7]These authors contributed equally: Jesse Hudspeth, Kai Rogge. ✉e-mail: sebastiaan.werten@i-med.ac.at

**Fig. 1 | Psilocybin biosynthesis.** The psilocybin pathway originates from L-tryptophan, which undergoes decarboxylation, oxygenation and phosphate ester formation to yield norbaeocystin, the acceptor substrate for the methyltransferase PsiM. SAM S-adenosyl-L-methionine, SAH S-adenosyl-L-homocysteine. A third methyl transfer by PsiM to yield aeruginascin has been postulated but lacks experimental evidence.

aeruginascin. Intriguingly, our data suggest a direct evolutionary relationship between PsiM and post-transcriptional regulators of the METTL16 family, a striking case of evolutionary tinkering most likely driven by substrate mimicry.

## Results

### The structure of PsiM at atomic resolution

To investigate the three-dimensional structure of PsiM, the recombinant transferase was crystallised in the presence of the methyl-depleted coenzyme (S-adenosyl-L-homocysteine or SAH) and norbaeocystin, the natural substrate. Two morphologically distinct crystal forms were obtained, one orthorhombic containing a single molecule in the asymmetric unit (ASU) and the other monoclinic containing two. Both crystal forms grew under identical conditions and often appeared within the same drop. The two structures (Fig. 2 and Supplementary Table 1) were determined at 0.91 Å and 1.18 Å resolution, respectively, revealing immediately recognisable electron density for coenzyme and substrate (Fig. 2c). Conformational differences (Supplementary Fig. 1) are mainly found in surface-exposed loops and in a short N-terminal section that originates from the expression vector. This region contains a TEV protease cleavage site that forms an α-helix in the orthorhombic crystal form, while it is most likely disordered in the monoclinic form.

Consistent with predictions[18,19], the structure of PsiM (Fig. 2a, b) comprises a Rossmann fold characteristic of class I SAM-dependent methyltransferases[23]. This fold consists of 6 parallel β-strands, interlaced with α-helices and followed by a single antiparallel strand. The resulting 7-stranded β-sheet exhibits the typical 3-2-1-4-5-7-6 topology (Fig. 2b). Other hallmarks of class I methyltransferases include a glycine-rich β-turn (residues 107–109, sequence GTG) that wraps around the coenzyme[24] and an acidic residue (Glu131) engaged in hydrogen bonds with both hydroxyl groups of the ribose moiety[25]. In addition to the Rossmann fold, PsiM features an N-terminal jaw-like domain also seen in some other methyltransferases[26], as well as unique N- and C-terminal extensions.

Inspection of the active site and the contacts between the canonical NPPF motif[23], SAH and norbaeocystin (Fig. 2c) provides immediate insight into key aspects of the catalytic mechanism. The amino group of norbaeocystin is held in place by two hydrogen bonds, one involving the backbone carbonyl of Pro184 (N-O distance: 2.8 Å) and the other with the side-chain oxygen of Asn183 (N-O distance: 2.9 Å). The geometry of these interactions matches the tetrahedral (sp³) coordination of the nitrogen and directs its lone pair towards the space where the methyl group of SAM would be located. Together with the polarisation caused by the hydrogen bonds, this arrangement poises the nitrogen atom for its nucleophilic attack on the methyl group. Consistent with a key role of the Asn183 side chain in the process, the N183A mutation renders PsiM catalytically inactive[18].

### Substrate binding and specificity

The open reading frame encoding PsiM in *Psilocybe* mushrooms is located within a dedicated gene cluster responsible for the production of psilocybin[10], suggesting that the enzyme may exclusively recognise norbaeocystin and baeocystin. Indeed, earlier work has indicated that closely related compounds such as 4-hydroxytryptamine and 4-hydroxy-L-tryptophan are not processed by PsiM[10]. These observations are explained by our structures as the binding pocket is found to completely envelop the substrate, interacting specifically with both the indole ring and the phosphate. The indole is sandwiched between Phe202 and Met217, while Leu68, Phe186 and His210 contribute additional hydrophobic contacts (Fig. 3). The side chain of Tyr187 forms a weak hydrogen bond with the indole nitrogen (O-N distance: 3.1 Å) and helps to position Phe186. This explains the earlier observation that a Y187A substitution renders PsiM catalytically inactive[18]. Arg75 (also known to be required for catalytic activity[18]) and Arg281 tightly grip the phosphate moiety of norbaeocystin and compensate its negative charge. Interestingly, the side chain of Asn183, whose oxygen atom forms a catalytically important hydrogen bond with the amino group of norbaeocystin, engages in two additional hydrogen bonds by means of its nitrogen. One of these involves the phosphate of the substrate (N-O distance: 2.9 Å), the other the carboxyl of SAH (N-O distance: 3.0 Å). Thus, coenzyme binding, substrate recognition and catalysis are directly linked by a single, pivotal residue.

Consistent with earlier predictions[18], the region following strand β4 (residues 189–221), which lacks homology to other methyltransferases, acts as a substrate recognition loop (SRL). The SRL forms a highly contorted structure, which is stabilised by the interaction with norbaeocystin and by intramolecular hydrogen bonds that facilitate turns. For instance, the side chain of Ser196, which is required for enzymatic activity[18], stabilises the conformation of the main chain by forming a hydrogen bond with the backbone amide of Ala198. The SRL not only assists in recognising the substrate, but also forms a barrier that physically closes off the binding pocket, locking the substrate in the interior of the enzyme. Consequently, part of the polypeptide chain must be able to make way to allow the substrate to enter the active site, as well as the reaction product to leave. To gain insight into this process, we crystallised the PsiM-SAH complex in the absence of substrate. A tetragonal crystal form was obtained that contained two PsiM molecules per ASU and diffracted to 2.5 Å (Supplementary Fig. 2 and Supplementary Table 1). Although structural differences with respect to the ternary complex are minimal, part of the SRL (residues 198–207 and 197–209 in chains A and B, respectively) is no longer observed in the electron density, suggesting disorder in the absence of the substrate. The result is a wide opening, sufficiently large to render the active site accessible to small organic molecules.

Efforts to obtain crystals of PsiM in its ligand-free state or in the presence of substrate alone were unsuccessful, suggesting that additional regions of the protein are disordered in the absence of the coenzyme. Isothermal titration calorimetry (ITC, Supplementary Fig. 3) reveals that SAH is able to bind to PsiM in the absence of any substrate (panel g, $K_d$: 66 ± 49 μM), while binding of norbaeocystin (panel c, $K_d$: 35.7 ± 5.4 μM) or baeocystin (panel d, $K_d$: 77.2 ± 3.5 μM) requires the presence of SAH (compare panels a and b to c and d, respectively). Taken together, our structural and biophysical data point towards a strictly sequential binding mechanism. First, PsiM interacts with the coenzyme, an event that creates or stabilises the

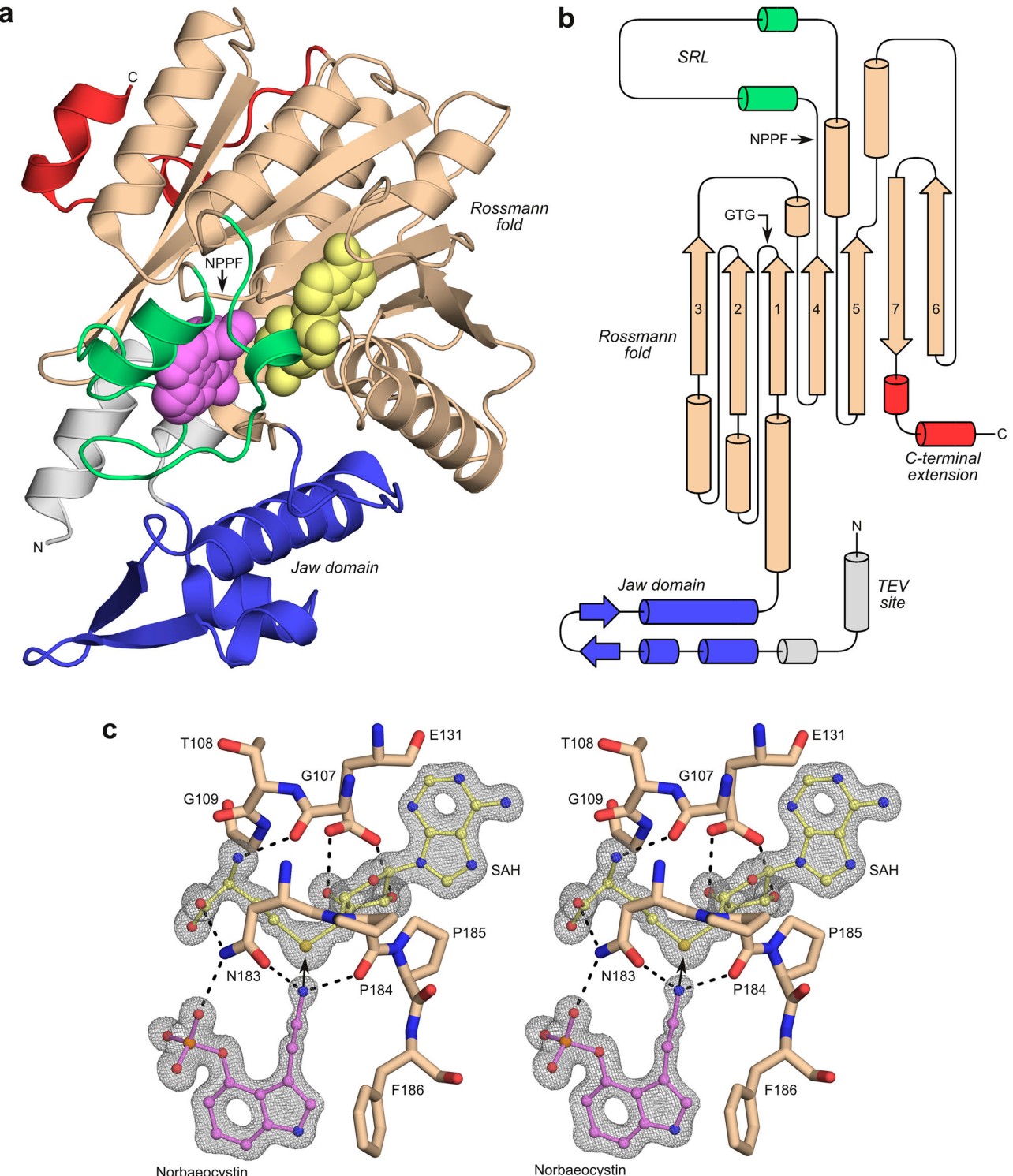

**Fig. 2 | The three-dimensional structure of PsiM in complex with SAH and norbaeocystin. a** Ribbon representation of the orthorhombic crystal structure at 0.91 Å resolution (PDB 8PB4). The Rossmann fold is depicted in beige, the N-terminal jaw-like domain in blue and two helices forming a unique C-terminal extension in red. The region 189–221, which bears no homology to other methyltransferases and acts as a substrate recognition loop (SRL), is coloured green. SAH and norbaeocystin are represented as yellow and pink space-filling models, respectively. The N-terminal extension preceding the jaw domain (shown in grey)

mainly consists of a TEV protease site originating from the expression vector. **b** Topology diagram for the PsiM fold (colours as in **a**). **c** Wall-eyed stereo figure showing OMIT density for the two ligands in the monoclinic crystal form, PDB 8PB3 ($2mF_o$-$DF_c$ with SAH and norbaeocystin omitted simultaneously, map contoured at 1.0 σ). Also shown are functionally relevant protein residues (stick models) and key hydrogen bonds (dashed lines). An arrow indicates the direction of the nucleophilic attack.

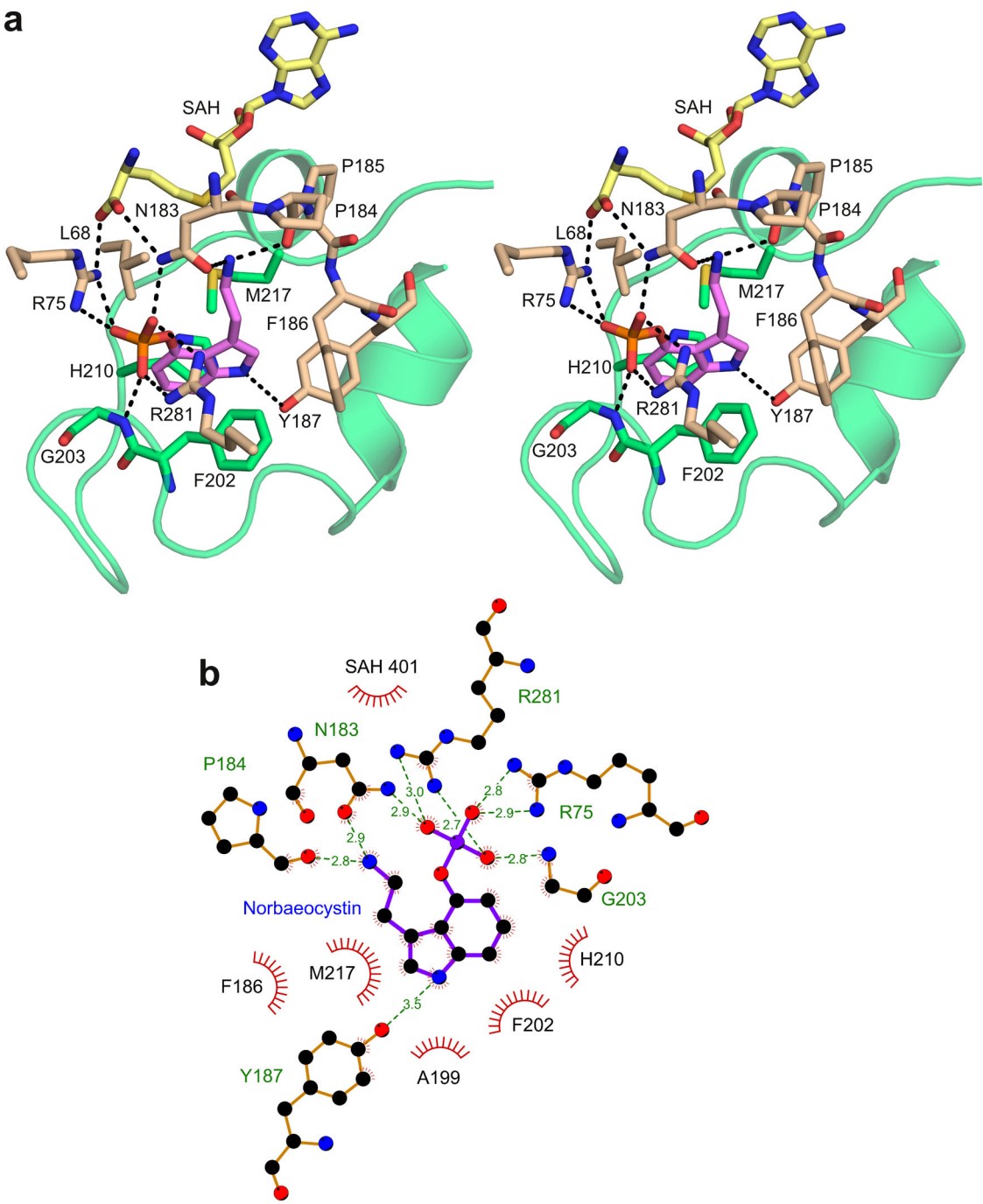

**Fig. 3 | Substrate recognition by PsiM. a** Wall-eyed stereo view of the substrate-binding pocket with SAH, norbaeocystin and protein side chains involved in ligand recognition represented as stick models. Hydrogen bonds are visualised as dashed lines. The substrate recognition loop is shown as a ribbon model. Colours as in Fig. 2. **b** Schematic plot showing PsiM-norbaeocystin contacts. Hydrogen bonds are shown as dashed lines, together with the corresponding bonding distances in Å.

active site. Thereupon the substrate binds, leading to stable folding of the SRL and closure of the pocket. This sequential model implies that exchange of SAH for SAM prior to the second round of methyl transfer is accompanied by the release of baeocystin, the intermediate product. Consistent with such a mechanism, free baeocystin is found to accumulate in enzymatic reactions in vitro (Fig. 4a).

**The reaction cycle**

The orthorhombic PsiM-SAH-norbaeocystin crystal structure at 0.91 Å (Fig. 5a) reveals two distinct substrate binding modes. The main structure (64% occupancy) is identical to the one seen in the lower-resolution monoclinic form. In the second binding mode (36%), the

indole ring is shifted away from its main position by 0.9 Å, providing space for the ethylamine chain to adopt an alternative conformation. This enables the amino group to form hydrogen bonds with the phosphate as well as the side chain of Arg281. The alternative state is incompatible with methyl transfer in the presence of SAM, but likely to favour deprotonation of the $NH_3^+$ group before the transition state forms.

To investigate the enzymatic reaction cycle in more detail we made use of sinefungin, a general methyltransferase inhibitor that mimics SAM. The 0.89 Å structure of the PsiM-sinefungin-norbaeocystin complex (Fig. 5b) provides insight into the state of the active site immediately preceding the first methyl transfer. In the

**a**

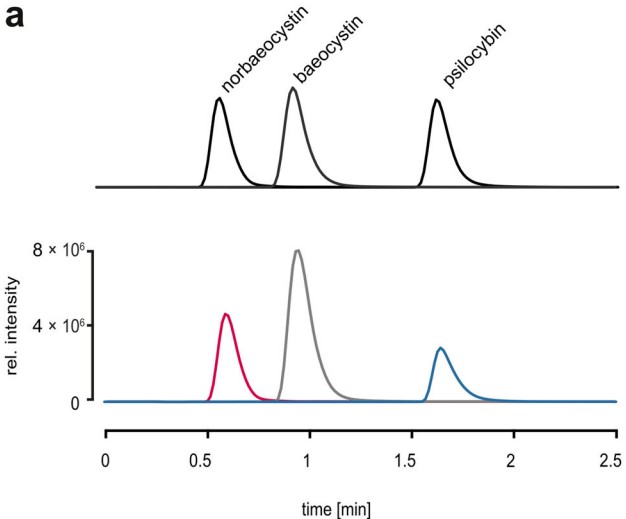

**b**

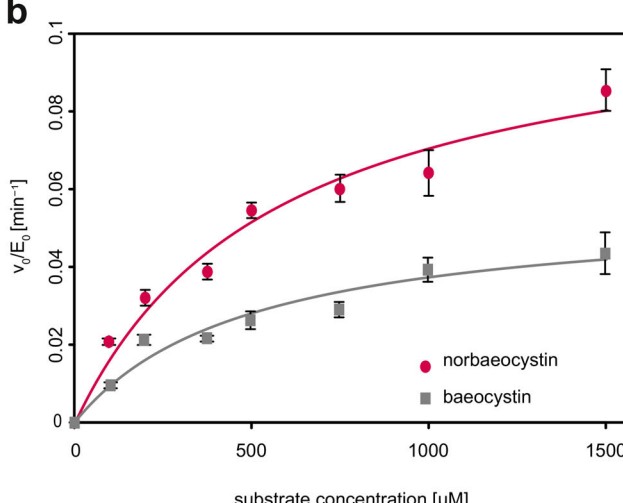

**Fig. 4 | Enzymatic activity of PsiM. a** In vitro activity assay monitored by means of single ion chromatograms, revealing accumulation of the monomethylated intermediate baeocystin. Upper lane: overlaid chromatograms of authentic standards. Bottom lane: a typical PsiM-catalysed methylation reaction after 2 h, represented by chromatograms for $m/z$ 257 [M + H]$^+$ (red, norbaeocystin), $m/z$ 271 [M + H]$^+$ (grey, baeocystin), and $m/z$ 285 [M + H]$^+$ (blue, psilocybin). **b** Michaelis-Menten kinetics of the first (red; $K_m = 575 \pm 100$ μM, $k_{cat} = 0.11 \pm 0.01$ min$^{-1}$) and the second methylation reaction (grey, $K_m = 492 \pm 154$ μM, $k_{cat} = 0.06 \pm 0.01$ min$^{-1}$). Data are presented as mean values of three independent experiments ($n = 3$), with error bars corresponding to the sample standard deviation (SD). Source data are provided in the supplementary Source Data file.

absence of a nucleophilic attack, the amino groups of norbaeocystin and sinefungin − the latter occupying the position of the methyl of SAM − are engaged in a hydrogen bond (N-N distance: 2.9 Å). In order to accommodate the length of the hydrogen bond, which exceeds that of a covalent sulfur-carbon bond by approximately 1.3 Å, sinefungin is forced into an energetically unfavourable conformer that is not observed with SAH. An alternative state (36% occupancy) is observed where sinefungin adopts a less constrained conformation, very similar to the one seen for SAH. This, however, forces the norbaeocystin amino group away from the active site, towards a different position that allows it to form an equivalent hydrogen bond with the inhibitor. The hydrogen bonds with Asn183 and Pro184 on the other hand are lost. In view of the similarity between sinefungin and SAM it is conceivable that an analogous dynamic equilibrium (involving CH-N instead of NH-N hydrogen bonds) exists in the PsiM-SAM-norbaeocystin complex, immediately after substrate binding and prior to the actual nucleophilic attack or the formation of a pre-transition state CH-N tetrel bond[27].

To characterise the state of the active site directly after the first methyl transfer, we cocrystallised PsiM with SAH and the monomethylated product, baeocystin. A well-defined 0.93 Å structure was obtained that revealed unambiguous electron density for both ligands (Fig. 5c). Interestingly, the monomethylated amino group of baeocystin appears to immediately retract from the transition state position to occupy a pair of alternative positions (with 60% and 40% occupancies). One of the alternative positions involves a movement of the indole ring away from the coenzyme, resulting in a C-S distance of 3.4 Å. This movement may constitute a first step towards product release.

The transient state immediately preceding the second methyl transfer was investigated by analysing a PsiM-sinefungin-baeocystin complex (Fig. 5d). In the resulting 0.92 Å structure, the methyl group of baeocystin points towards the side chain of Asn183, which creates the necessary space by rotating away from the catalytic site. This leads to a 1.3 Å displacement of the Asn183 oxygen atom compared to the complex with sinefungin and norbaeocystin, where the oxygen is engaged in a hydrogen bond with the substrate amine. The only conceivable alternative, a conformation with the methyl group pointing towards the carbonyl of Pro185, is not observed at all, presumably

because the constrained backbone of the NPPF motif cannot move aside. Intriguingly, we find that substrate occupancy is only 61% in this complex. A large portion of the SRL (residues 198–208), Arg281, as well as numerous associated water molecules also show partial occupancy, consistent with substrate-dependent folding. As all ternary complexes were crystallised in the exact same manner, these observations strongly suggest that, in the presence of the SAM analogue sinefungin, the affinity of PsiM for baeocystin is lower than that for norbaeocystin. This difference in affinity was confirmed by ITC experiments (Supplementary Fig. 3). Furthermore, chromatographic analyses of typical in vitro assays demonstrate that the monomethylated intermediate baeocystin accumulates and becomes the dominant species before psilocybin formation sets in (Fig. 4a). Results of quantitative experiments (Fig. 4b) can be explained by Michaelis-Menten kinetics and indicate that the first methyl transfer is approximately twice as fast as the second, with $k_{cat}$ values of $0.11 \pm 0.01$ min$^{-1}$ and $0.06 \pm 0.01$ min$^{-1}$. The Michaelis constants ($K_m$) for norbaeocystin and baeocystin are $575 \pm 100$ μM and $492 \pm 154$ μM. These values translate into catalytic efficiencies of 0.191 and 0.122 nM min$^{-1}$ for norbaeocystin and baeocystin, respectively. Taken together, these observations demonstrate that PsiM is a comparatively poor dimethyltransferase.

We also crystallised PsiM in the presence of the final reaction products, SAH and psilocybin. The corresponding 0.94 Å structure reveals two conformations of the hallucinogen, which only differ in the orientation of the dimethylated amine (Fig. 5e). One conformation (46% occupancy) likely reflects the situation following the second transfer reaction, with the methyl carbon at 3.1 Å from the sulfur of the coenzyme and the nitrogen maintaining its hydrogen bond to the carbonyl of Pro184. In the other conformation (54% occupancy), the dimethylated amine is rotated by approximately 180°, allowing the nitrogen, now facing away from the carbonyl, to form an intramolecular hydrogen bond with the phosphate of psilocybin. This state represents a likely first step towards the release of psilocybin from the enzyme.

Superposition of the five atomic-resolution crystal structures reveals that the protein models are nearly identical, with pairwise Cα-RMSD values between 0.10 Å and 0.32 Å (Fig. 5f). In contrast, the positioning of the ligand with respect to the substrate-binding cavity varies considerably. By comparing all of the crystal structures and

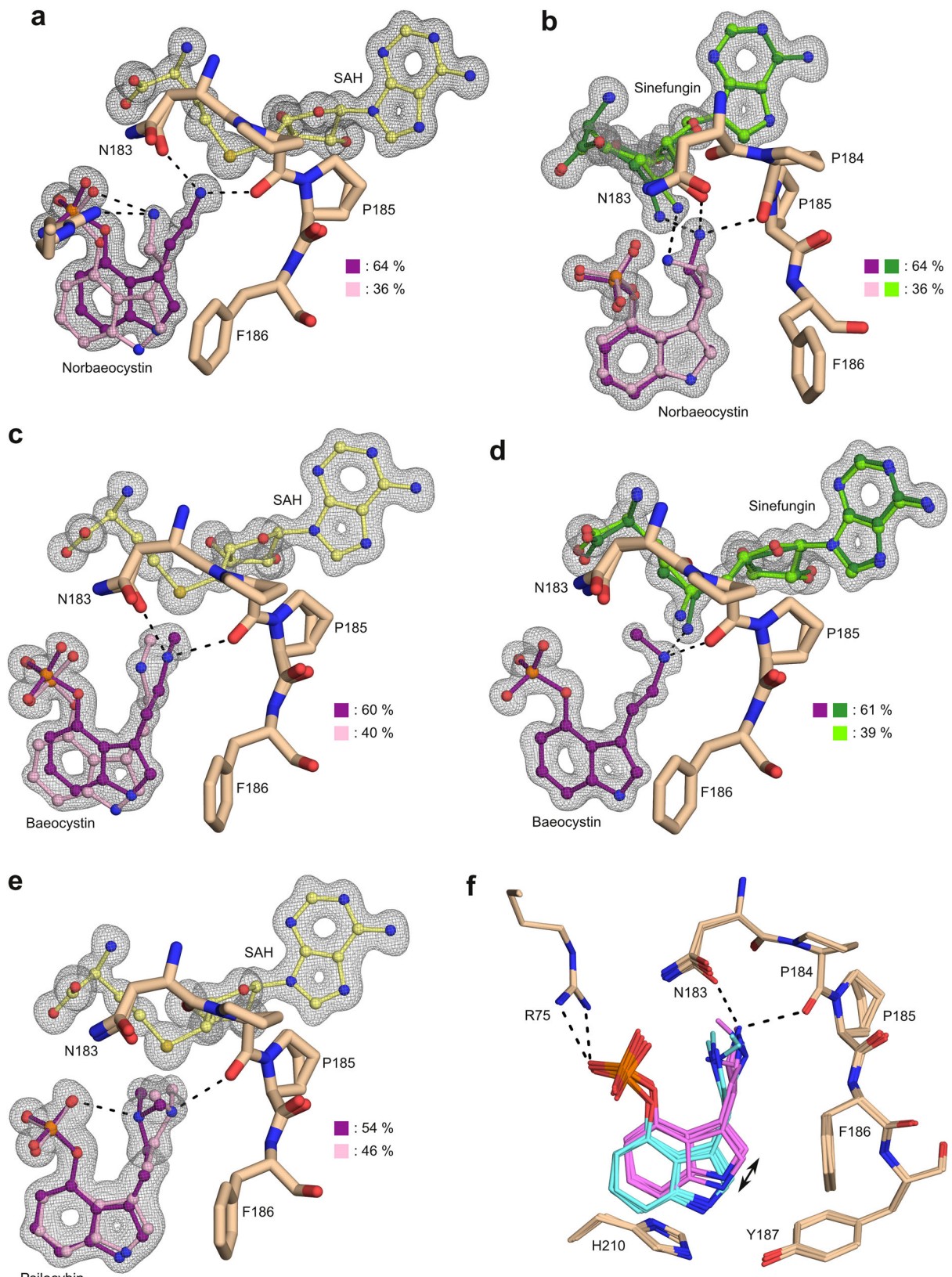

alternative conformers, two principal binding modes can be distinguished, which are related by a rotation of the ligand around the phosphate moiety, roughly within the plane of the indole ring (Fig. 5f). In one of the binding modes, the ethylamide nitrogen is positioned at the catalytic centre and hydrogen bonded to the Pro184 carbonyl and, in the case of norbaeocystin, to the side chain of Asn183. The second binding mode corresponds to a disengaged state with the indole ring positioned further from the catalytic centre, providing space for the ethylamine chain to adopt a variety of conformations depending on available contacts. The second binding mode is observed in both post-reaction complexes (i.e. in the presence of methylated products and SAH) and in the complex containing SAH and norbaeocystin. We

**Fig. 5 | Consecutive stages of the reaction cycle.** The NPPF motif of PsiM is shown as a stick model (beige). SAH (yellow), sinefungin (light/dark green) and (nor) baeocystin/psilocybin (pink/purple) are represented as ball-and-stick models, with the darker colour representing the main conformation. Corresponding OMIT maps are shown in grey ($2mF_o\text{-}DF_c$ with both ligands omitted from the model, maps contoured at 1.0 σ). Key hydrogen bonds are shown as dashed lines. Each model is rotated slightly differently so as to clearly reveal its relevant properties. **a** The PsiM-SAH-norbaeocystin complex (PDB 8PB4, 0.91 Å), revealing an alternative substrate conformation potentially involved in amine deprotonation. **b** The PsiM-sinefungin-norbaeocystin complex (PDB 8PB5, 0.89 Å), a model for the state of the complex prior to the first methyl transfer. **c** The PsiM-SAH-baeocystin complex (PDB 8PB6, 0.93 Å), revealing the state of the active site immediately after the first methyl transfer. **d** The PsiM-sinefungin-baeocystin complex (PDB 8PB7, 0.92 Å), approximating the situation immediately before the second methyl transfer. **e** The PsiM-SAH-psilocybin complex (PDB 8QXQ, 0.94 Å). **f** Superposition of the structures shown in panels a–e. Interactions shown as dashed lines belong to the main conformation of the PsiM-SAH-norbaeocystin complex (**a**). Two distinct ligand positions relative to the binding site have been emphasised by means of the colours magenta and cyan.

propose that the disengaged state provides an exit route for product molecules, as well as an initial substrate binding mode that favours deprotonation of the ethylamide before the transition state is formed.

## PsiM evolved from epitranscriptomic regulators

Analysis of our PsiM structures using the programme Dali[28,29] indicates that the most similar entries found in the protein data bank (PDB) correspond to the catalytic domain of the human m[6]A-methyltransferase METTL16, with Dali $Z$-scores ranging from 25 to 35. The highest-scoring entry (PDB 6DU4) can be superimposed onto PsiM with a $C_\alpha$-RMSD of 1.5 Å for 260 $C_\alpha$-atoms (Fig. 6a). METTL16 is involved in post-transcriptional gene regulation and methylates specific sites in mRNA as well as in snRNA[26]. Interestingly, the protein not only features a highly similar Rossmann fold with an identical 3-2−1-4-5-7-6 topology, but also possesses the N-terminal jaw domain seen in PsiM. This distinctive structural motif, consisting of three α-helices and two β-strands, plays a crucial role in RNA-recognition by METTL16[30,31]. BLAST[32] searches at NCBI[33] only identified the motif in METTL16 itself and in the closely related m[6]A-methyltransferases of the RlmF family, which modify ribosomal 23 S RNA[34,35]. Although these observations suggest a close evolutionary relationship between PsiM and the METTL16/RlmF group of m[6]A-methyltransferases, most residues that are critical for nucleic acid binding have not been conserved in the fungal enzyme.

Results of a sequence-based phylogenetic analysis are shown in Fig. 6b and Supplementary Fig. 4. Consistent with the similarity that was detected at the structural level, the methyltransferases most closely related to PsiM belong to the METTL16 family, which itself branches off from the more ancient ribosomal 23 S m[6]A-methyltransferases. These findings seem highly surprising, given that the ancestral transferases are regulatory proteins, invariably act on RNA and, perhaps most intriguingly, only transfer a single methyl group to their target amine, whereas PsiM dimethylates its substrate. A likely explanation for the unexpected evolutionary relationship between PsiM and METTL16 is provided in Fig. 6c, which shows a superposition of the substrate-binding pockets of the PsiM-SAH-norbaeocystin and METTL16-mRNA complexes. The interaction of PsiM with its phosphorylated substrate norbaeocystin shows a striking resemblance to the binding of METTL16 to the target RNA nucleotide, with the phosphate groups occupying nearly identical positions and the indole ring mimicking the adenine base. Thus, substrate mimicry presumably explains how PsiM arose from m[6]A-monomethyltransferases through a process of evolutionary tinkering[36].

## A single amino acid substitution in PsiM conferred dimethyl-transferase activity

The ancestral enzymes that gave rise to PsiM exclusively mediate monomethylation of the target adenosines in their RNA substrates, raising the question how PsiM gained the ability to catalyse dimethylation. Our structures show that productive binding of the monomethylated intermediate baeocystin to the catalytic site of PsiM requires the Asn183 side chain to move aside by approximately 1 Å, generating the necessary space to accommodate the methyl group that points in its direction. Interestingly, the rotational movement of the side chain is made possible by the fact that unoccupied space remains towards Asn247, the residue located behind Asn183 (Fig. 7a). Asn247 is strictly conserved in PsiM (Fig. 7b) and forms a hydrogen bond with Asn183 as soon as the latter adopts its alternative conformation to make way for the substrate. In contrast, sequence alignments show that in m[6]A-methyltransferases the corresponding residue is either Leu or Met − bulky hydrophobic residues unable to engage in an NH-N hydrogen bond − whereas surrounding residues are conserved with respect to PsiM. This suggests that steric hindrance restricts the mobility of Asn183 in these enzymes, while the (M/L)247 N substitution in PsiM constitutes the evolutionary adaptation enabling dimethylation. Superposition of PsiM onto METTL16 (Fig. 7a) corroborates the idea that steric hindrance prevents movement of Asn184 in the latter. To further test our hypothesis, we investigated the effect of the reverse mutation, N247M, in PsiM. Results (Fig. 7c) indicate that the mutated enzyme retains its ability to methylate norbaeocystin to baeocystin, but fails to produce significant amounts of psilocybin, the dimethylated product.

## PsiM is unable to catalyse trimethylation to aeruginascin

Various authors describe aeruginascin (Fig. 1), the quaternary ammonium derivative of psilocybin, as a mushroom metabolite[20–22,37]. However, an inconsistent picture has emerged as to whether aeruginascin is a product of the psilocybin biosynthetic pathway. While the trimethylated compound was detected in transgenic psilocybin-producing yeast[16], it was not present in engineered *E. coli* and *A. nidulans* strains expressing the *psi* genes[14,15]. In vitro assays did not yield aeruginascin either[10]. We sought to confirm PsiM's apparent incapacity for a third *N*-methyltransfer by in vitro biochemical assays. First, optimal reaction conditions were established. Using the first methylation reaction as readout, optimum turnover was found in a window between 20−30 °C and at a pH 8.4, which may favour a third methylation, analogous to the trimethylating L-lysine methyltransferase of *Neurospora crassa*[38]. Subsequently, we performed comparative in vitro assays to monitor methylation of norbaeocystin, baeocystin, and psilocybin by PsiM (Supplementary Fig. 5). When norbaeocystin or baeocystin were added as acceptor substrates to PsiM, psilocybin formation was observed as expected. In contrast, none of the reactions led to detectable amounts of aeruginascin. The inability of PsiM to perform trimethylation most likely indicates that the proline-rich NPPF motif lacks sufficient flexibility to accommodate a dimethylated substrate, which would require main-chain conformational changes and a shift of the Pro184 carbonyl.

## Discussion

In the present study, we describe the reaction mechanism behind the biosynthesis of psilocybin and present the highest-resolution crystal structures for any methyltransferase to date. The structures are consistent with the "proximity and desolvation" model[26,39,40], which states that the active site of a methyltransferase forces the reactive groups into close proximity while excluding any interfering solvent molecules from their environment. Formation of so-called near-attack

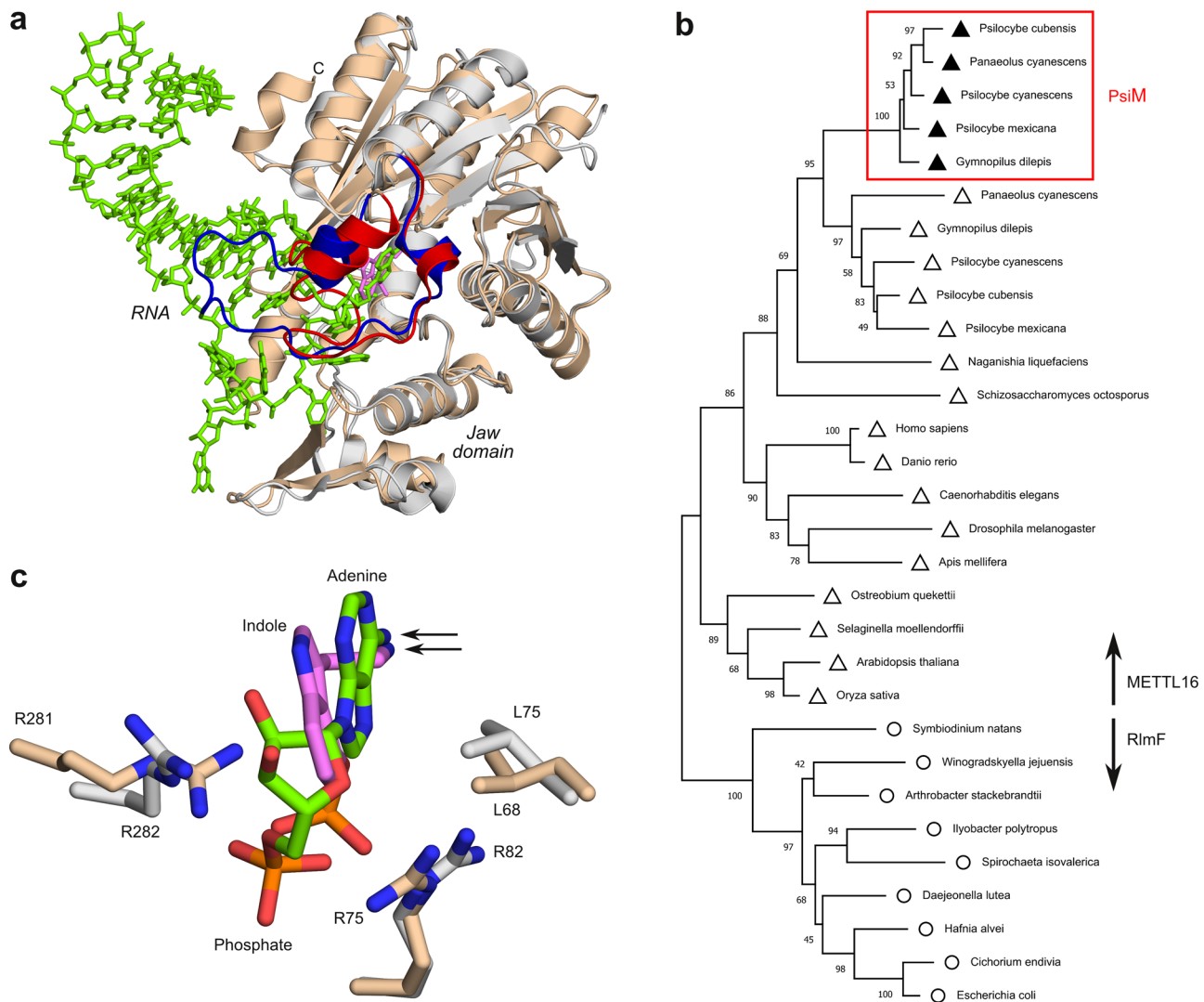

**Fig. 6 | PsiM evolved from epitranscriptomic m⁶A-methyltransferases.**
**a** Superposition of PsiM (beige) and a METTL16-RNA complex (PDB 6DU4, protein grey, RNA green). The substrate recognition loop of PsiM is coloured red, the equivalent RNA-binding loop of METTL16 blue. **b** Maximum-likelihood (ML) phylogenetic analysis of PsiM and related m⁶A-methyltransferases. Circles denote RlmF, open triangles METTL16 and filled triangles PsiM. The percentage of trees in which the associated sequences clustered together in a bootstrap analysis is shown next to the branches. **c** Enlarged view of the superposed substrate-binding pockets in **a**, with norbaeocystin (pink), the METTL16 target nucleotide (green) and conserved residues lining the binding site (beige and grey) as stick models. The amines that are methylated by the two enzymes are indicated by arrows.

conformers (NACs)[41–43], electronic pre-organisation[44,45], cratic effects[46,47] and compaction[48–50] have often been described as likely contributing factors, whereas more recent studies highlight the role of pre-transition state tetrel bond formation[27], chalcogen bonding[51] and CH hydrogen bonding[52] in the process. Although the (pre-)transition state cannot be directly observed in structural studies – the ternary complexes under investigation do not contain SAM as it would allow the reaction to proceed – our atomic-resolution data prompt a detailed analysis of the active site geometry in the presence of two closely related coenzyme analogues. Modelling SAM on the basis of the experimentally determined coordinates of SAH or sinefungin places the transferable methyl group at 1.7–1.8 Å from the fully engaged nucleophilic nitrogen, *i.e.* equidistant from the nitrogen and the sulfur, with S-C-N angles of 167° and 173°, respectively (Fig. 8). Thus, the ligand binding mode observed in our structures would be expected to stabilise the transition state and favour shortening of the initial tetrel bond between the nucleophile and the methyl carbon[27]. Consistent with observations made for other methyltransferases[52], several CH-hydrogen bonds link the coenzyme to PsiM (Fig. 8). On the other hand,

we do not observe any chalcogen bonding[51] interactions between PsiM and the sulfur atom of the coenzyme.

Intriguingly, the atomic-resolution structures reveal a number of alternative substrate and product binding modes. Roughly, two types of ligand positioning with respect to the binding site can be distinguished. In the first, the substrate is fully engaged and the nucleophilic nitrogen held in place by hydrogen bonds. In the second, the indole ring is further away from the catalytic site, providing space for the ethylamide to change its conformation and engage in a variety of alternative interactions. This disengaged state is incompatible with methyl transfer, but likely to play a role during initial substrate capture, subsequent deprotonation of the amine and, following the reaction, as a first step towards product release.

Our study also sheds a new light on the evolution of SAM-dependent methyltransferases and particularly on their ability to adapt to novel substrates. Consistent with the idea that highly variable loop regions inserted into the core Rossmann fold play a crucial role in this process, we find that the SRL of PsiM – originally a motif interacting with stem-loop structures in RNA – developed into a highly specialised

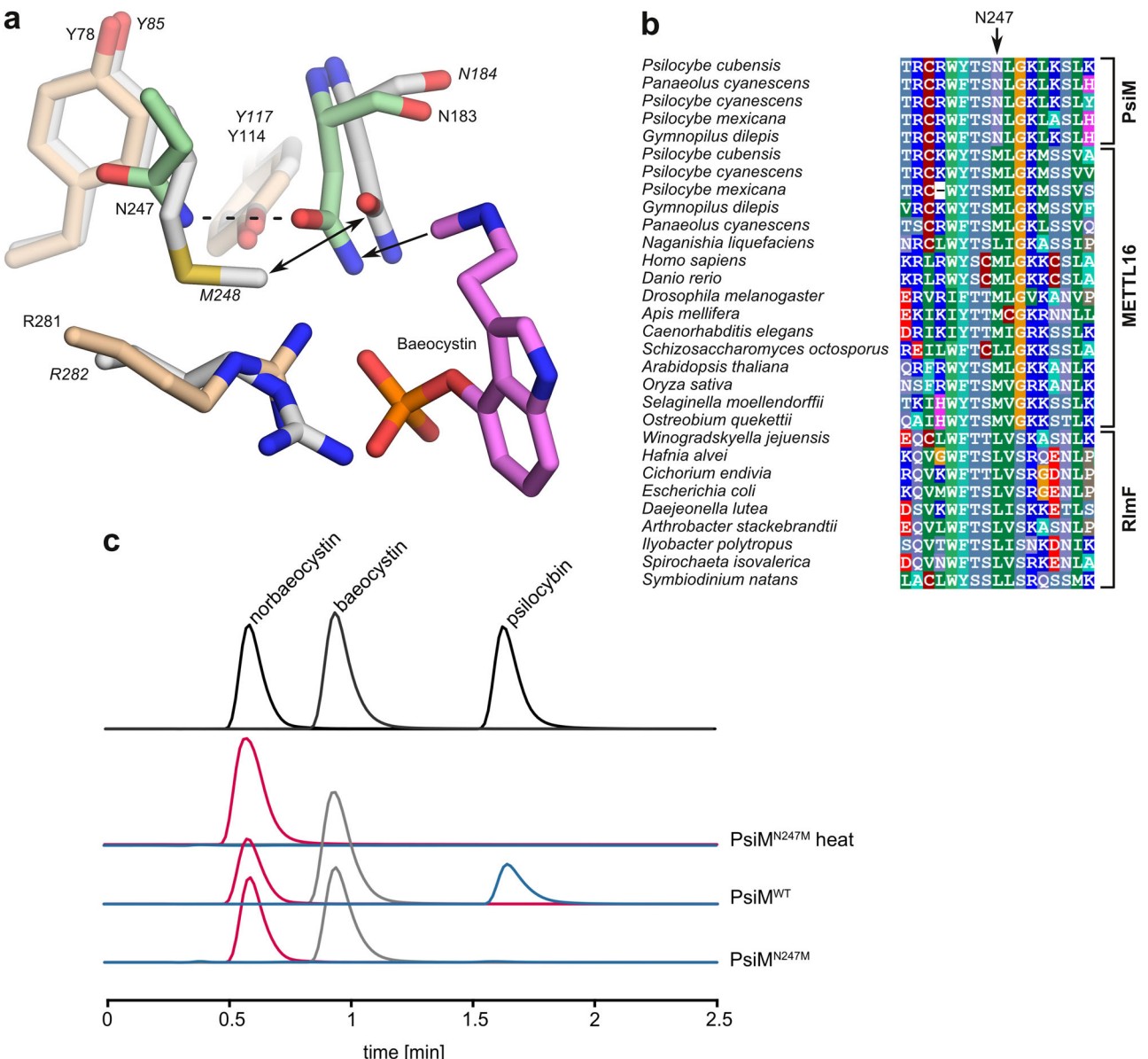

**Fig. 7 | A single amino acid substitution holds the key to dimethylation ability.** **a** Superposition of PsiM (beige with key residues Asn183 and Asn247 in green) and the METTL16-SAH complex (chain A of PDB 6GFK, grey). Residue numbers are shown in normal print and italics, respectively. The methyl group in baeocystin forces Asn183 of PsiM to shift to the left (arrow). Asn247 provides the necessary space and in addition stabilises the alternative conformation of Asn183 through a hydrogen bond (dashed line). A comparable movement of the corresponding residue in METTL16 (Asn184) is prevented by the presence of Met248 (double arrow) in the otherwise strictly conserved protein core. **b** Alignment of the region around N247 in PsiM with the sequences of other METTL16- and RlmF-like transferases. **c** Experimentally determined (di)methylation activity of wild-type PsiM and the N247M mutant. Shown are single ion chromatograms of norbaeocystin (red chromatogram, $m/z$ 257 [M + H]⁺), baeocystin (grey, $m/z$ 271 [M + H]⁺) and psilocybin (blue, $m/z$ 285 [M + H]⁺). Top lane: overlaid chromatograms of authentic standards. Second lane: negative control with heat-denatured enzyme.

lid structure that recognises the small-molecule substrate and sequesters it by closing off the binding site (Fig. 6a). In contrast, the Rossmann fold proper, the substrate-binding pocket and the catalytic centre remained virtually unchanged in the course of evolution. Emergence of PsiM from m⁶A-methyltransferases appears to have been driven by substrate mimicry, ultimately resulting in a moderately efficient small-molecule dimethyltransferase. Although the presence of N247 in PsiM, instead of M274 as in METTL16, enables PsiM to catalyse dimethylation, the second methyl transfer remains less efficient than the first. Moreover, exchange of SAH for SAM requires release of the intermediate product, baeocystin, further hampering the second round of methylation and favouring accumulation of the monomethylated species. This may indicate that the enzyme is caught in local optimum, an "evolutionary cul-de-sac", as a consequence of the inherent limitations of its original scaffold. Consistent with this idea, small-molecule methyltransferases that are known to efficiently di- or trimethylate amines, such as the histidine methyltransferase EgtD[53,54], the *N,N*-8-demethyl-8-amino-D-riboflavin methyltransferase RosA[55] and the phosphoethanolamine methyltransferase PfPMT[56], possess an altogether different active site architecture without an NPPF motif and are capable of exchanging the coenzyme while the substrate remains bound. With the notable exception of PsiM, the NPPF motif has only been found in N-methyltransferases acting on planar substrates where the target amine is part of a conjugated system[23]. It thus seems plausible that the rigid geometry of the proline-rich sequence helps imposing sp³ hybridisation, rather than sp², while guiding the lone pair

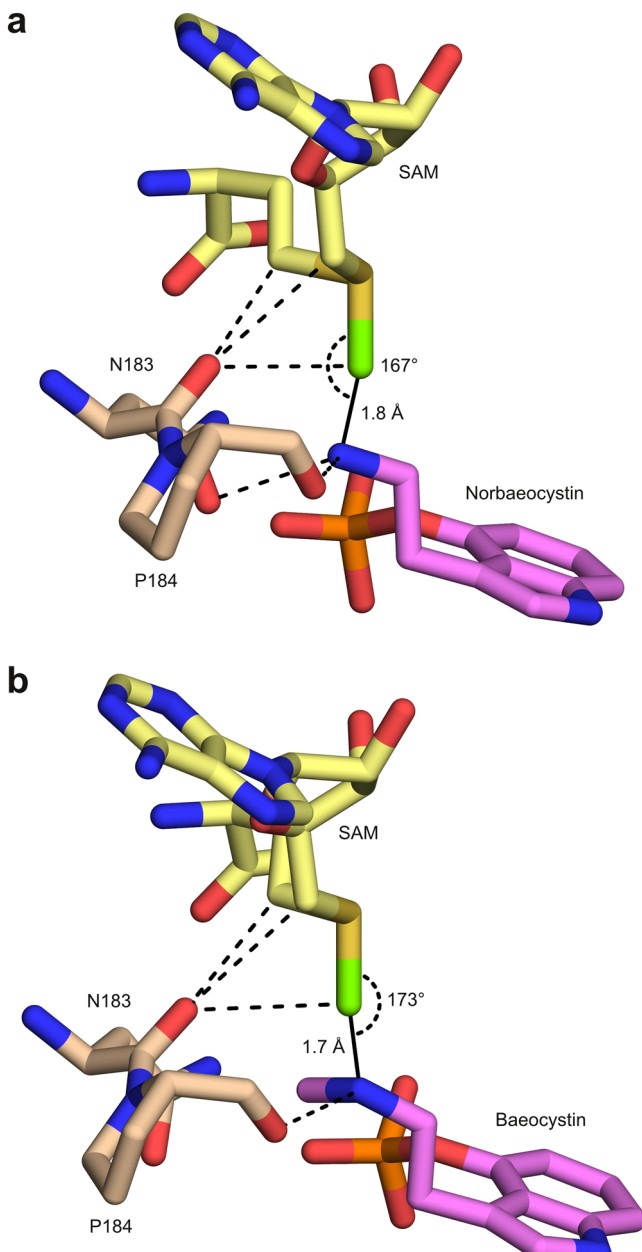

**Fig. 8 | Models of the active site in the presence of SAM.** Key PsiM residues are shown in beige, (nor)baeocystin in pink. Atomic coordinates for SAM (yellow) were generated on the basis of the corresponding high-resolution structures with SAH or sinefungin. The transferable methyl group is shown in bright green, with a solid line representing the nascent covalent bond. Hydrogen bonds positioning the nucleophilic nitrogen atom of the substrate are shown as dashed lines. Also shown are CH-O hydrogen bonds between the coenzyme and the carbonyl oxygen of Asn183. **a** Model of the PsiM-SAM-norbaeocystin complex, based on the experimentally determined structure of the corresponding complex with SAH. **b** Model of the PsiM-SAM-baeocystin complex, based on the experimentally determined structure of the corresponding complex with sinefungin.

towards the methyl group of SAM. The fact that the amines in the psilocybin precursors norbaeocystin and baeocystin are not part of a conjugated system and, in principle, should not benefit from such a mechanism, is a further indication pointing at the evolutionary origins of PsiM.

Reactive oligomers resulting from oxidation of the phenolic hydroxyl group of psilocin, the product of spontaneous dephosphorylation of psilocybin, precipitate proteins and cause intracellular

damage[57]. Hallucinogenic mushrooms protect themselves by means of another enzyme of the psilocybin biosynthetic pathway, the promiscuous kinase PsiK, which re-phosphorylates psilocin to psilocybin[13]. Formation of the tertiary amine is advantageous to the fungus as it enhances intramolecular hydrogen bonding between the phenolic hydroxyl group of the dephosphorylated species and the ethylamine[58]. This effect markedly slows down oxidation, enabling PsiK to keep pace with the dephosphorylation process[58]. In the light of these considerations, PsiM's incapacity to perform a third methylation seems consistent with the absence of protective hydrogen bond formation in aeruginascin's dephosphorylated degradation product, 4-hydroxy-*N,N,N*-trimethyltryptamine.

The METTL16 family of methyltransferases was recently discovered and shown to represent a novel class of epitranscriptomic "writer" proteins, which place regulatory m6A marks on mRNA[26,59]. METTL16 was also found to modify various kinds of non-coding RNA, including U6 snRNA. In humans, aberrant RNA modification patterns are frequently associated with disease, leading to significant interest in the emerging field of epitranscriptomics[60]. PsiM and the methyltransferase domain of human METTL16 are closely related and retain 37% sequence identity. Consequently, our results are likely to provide direct insight into the catalytic mechanism of METTL16. While the latter could be crystallised in its apo-form[61] and in complexes containing either SAH[31,61] or RNA[30], cocrystals with both ligands remain elusive[30]. Our structures of PsiM can be used as templates to model the corresponding ternary complexes of METTL16. Moreover, in view of the resemblance between the substrate-binding modes of the two enzymes, norbaeocystin and baeocystin may turn out to be effective lead compounds for the design of therapeutic METTL16 inhibitors[60], or indeed of novel general-purpose nucleotide analogues. Despite the remarkable level of similarity between PsiM and METTL16, several intriguing differences are also apparent. In the first place, the enzymatic activity of METTL16 is tightly controlled[26,30], consistent with its role in epitranscriptomic gene regulation. Residues Lys163 and Met167 of the so-called K-loop, which play essential roles in the regulatory mechanism, have not been conserved in PsiM (Supplementary Fig. 4). Furthermore, all METTL16 proteins carry variable and partially disordered C-terminal extensions, comprising a pair of structured "vertebrate conserved region" (VCR) domains that influence RNA-binding[61–63]. The VCR domains were also reported to mediate dimerisation[61], although this has been contested more recently[26]. Finally, METTL16 was shown to bind RNA before it recruits SAM[63], whereas our results indicate that PsiM must recruit its cofactor before it is able to interact with substrates. It is conceivable that initial binding of RNA to the METTL16 surface alone[30] triggers SAM recruitment. The presence of the coenzyme might then stabilise the active site and enable it to sequester the target nucleotide.

Accurate three-dimensional protein structures are key assets in bioengineering, enabling rational modification of enzymes and their substrates. Inspection of the substrate-binding pocket of PsiM reveals that moderately sized substituents at positions 6 and 7 of the indole ring are likely to be tolerated by the wild-type enzyme. Substitutions at the remaining ring positions, on the other hand, are likely to require substantial remodelling of the active site through mutagenesis. The phosphate moiety seems especially important for substrate recognition and presumably cannot be deleted or moved to a different position on the ring without severely affecting binding affinity. These structure-based findings are confirmed by recent experiments involving alternative substrates in psilocybin-producing *E. coli* strains[64]. As the relevance of psilocybin as a future medication against a growing number of common mental health conditions is becoming clear[7–9,65], we expect that the atomic-resolution structures of PsiM will constitute important tools in bioengineering efforts aimed at producing novel analogues with improved therapeutic properties.

## Methods

### Heterologous production and purification of PsiM for structural studies

The *psiM* gene of *P. cubensis* (GenBank entry ASU62238.1) was amplified with primers oJF105 and oJF47 using the plasmid pFB13 as a template[10]. The PCR product was cloned into pET-28a (Novagen) via the NheI and XhoI restriction sites. The resulting expression vector, pJF45, which was verified by sequencing, encodes full-length PsiM preceded by a hexahistidine tag and recognition sites for both thrombin and TEV protease. *E. coli* BL21 (DE3) was transformed with this vector and precultures grown overnight from single colonies at 37 °C in LB medium supplemented with 50 μg/ml kanamycin and 1% glucose. Expression cultures, prepared by diluting precultures 100-fold in fresh LB medium containing 50 μg/ml kanamycin and 1% glucose, were incubated at 37 °C until they reached an OD (600 nm) of 0.45. IPTG was added to a final concentration of 1 mM, followed by a further 3 h incubation at 37 °C. Upon centrifugation (60 min at 10,000 × *g*, 4 °C), cells from 1.2 l of culture were resuspended in 70 ml lysis buffer (20 mM Tris/HCl pH 8.0, 400 mM NaCl, 0.5 mM DTT) and disrupted by sonication on ice using a Sonopuls HD 2070 (Bandelin Electronic GmbH, Berlin, Germany). Insoluble cell debris was removed by centrifugation for 1 h at 40,000 × *g* and 4 °C. The cleared lysate was passed through a 0.45 μm filter (Whatman) and applied to a 5 ml metal affinity chromatography Ni$^{2+}$-cartridge (GE Healthcare). Following extensive washing, PsiM was eluted using a 0–300 mM imidazole gradient in lysis buffer and further purified by gel filtration (Superdex 200, GE Healthcare GmbH, Munich, Germany) in a buffer containing 20 mM Tris/HCl pH 8.0, 400 mM NaCl and 1 mM DTT. The protein appeared as a single band on Coomassie-stained SDS-PAGE gels at this stage. To proteolytically remove the hexahistidine-tag, the protein solution (10 ml total volume, 10 mg/ml) was dialysed for 12 h at 4 °C in a G2 Slide-A-Lizer (10 kDa MWCO, ThermoFisher Scientific) against 2 l digestion buffer (20 mM Tris/HCl pH 8.4, 150 mM NaCl, 2.5 mM CaCl$_2$), followed by the addition of 5 units of thrombin (Sigma) directly into the Slide-A-Lizer and continued dialysis for 12 h at 22 °C against 2 l of fresh digestion buffer. The protein was then dialysed for 12 h at 4 °C against 2 l protein storage buffer (20 mM Tris/HCl pH 8.0, 1 mM DTT) and concentrated to 31 mg/ml (0.87 mM) using a Vivaspin 20 ultrafiltration unit (10 kDa MWCO, Sartorius Lab Instruments GmbH, Göttingen, Germany). Protein concentration was determined spectroscopically, using a sequence-based estimate of the molar absorption coefficient at 280 nm (39400 M$^{-1}$ cm$^{-1}$). A PsiM·SAH complex was formed by adding a 2-fold molar excess of the coenzyme (Sigma). PsiM·SAH·(nor)baeocystin and PsiM·sinefungin·(nor)baeocystin complexes were prepared by adding a 2-fold molar excess of either the coenzyme or the inhibitor and a 5-fold molar excess of substrate (Usona Institute, Madison, WI, USA). Aliquots of the protein and the protein-ligand mixtures were flash-frozen in liquid nitrogen and stored at −80 °C until further use.

### X-ray crystallography

The PsiM·SAH·norbaeocystin complex was crystallised at 4 °C using the hanging drop method. Drops were produced by mixing equal volumes of the protein solution and the reservoir buffer, which contained 100 mM Tris/HCl pH 8.5, 20% PEG 8000 and 200 mM MgCl$_2$. After a week, morphologically distinct orthorhombic ($P2_12_12_1$) and monoclinic (*C*2) crystals began to appear, frequently within the same drop. The PsiM·SAH·baeocystin, PsiM·sinefungin·(nor)baeocystin and PsiM·SAH·psilocybin complexes were produced using the same approach, but cross-seeding was used to ensure growth of the better diffracting orthorhombic form. The PsiM·SAH complex was crystallised using the sitting drop method and a reservoir solution containing 100 mM bis-Tris propane pH 7.0, 30% PEG 300 and 15% PEG 1000. Tetragonal (*P*4$_3$) crystals of this complex appeared after several weeks.

Crystals were directly flash-cooled in liquid nitrogen, without the addition of further cryoprotectants. For X-ray diffraction experiments, a temperature of 100 K was used. All datasets were collected at ESRF ID23-1[66] and processed using xia2[67] with DIALS[68,69], with the exception of the data for the PsiM·SAH complex, which were collected at ESRF ID23-2[70] and processed using XDS[71,72]. Structures were solved by molecular replacement using Phaser[73], in combination with the CCP4 suite[74]. A poly-Ala model generated from PDB entry 6B92 was used as the search model to solve the orthorhombic crystal form of the PsiM·SAH·norbaeocystin complex. The refined structure was subsequently used as a search model to solve all other crystal forms. Coot[75] and Phenix[76] were used for iterative rounds of model building and refinement. Data collection and refinement statistics can be found in Supplementary Table 1.

All molecular graphics illustrations were produced with PyMOL 0.99rc6 (DeLano Scientific LLC, Palo Alto, CA, USA). The schematic overview of protein-ligand interactions shown in Fig. 3b was prepared using LIGPLOT[77]. Structural comparisons were performed using the SSM superposition method implemented in Coot[75].

### Production of PsiM for binding studies and activity assays

To produce hexahistidine-tagged PsiM without thrombin and TEV protease sites, *E. coli* KRX (Promega) was transformed with pFB13[10]. A 5 ml culture was grown overnight at 37 °C in LB medium containing 50 μg/ml kanamycin and then used to inoculate 500 ml of 2× YT medium with 50 μg/ml kanamycin. This culture was incubated at 37 °C until an OD (600 nm) of 0.6 was reached. Upon addition of L-rhamnose to a final concentration of 0.1%, the culture was transferred to 16 °C and incubated for another 20 h. The biomass was collected by centrifugation at 4000 × *g* and 4 °C, resuspended in 7.5 ml lysis buffer (50 mM phosphate buffer, 500 mM NaCl, 20 mM imidazole, pH 8) and lysed using a Sonopuls GM 3200 sonicator (Bandelin Electronic GmbH, Berlin, Germany). To remove cell debris, the lysate was centrifuged (10,000 × *g*, 20 min, 4 °C). The cleared supernatant was passed through a 0.45 μm filter (Whatman) and purified using a 1 ml HisTrap HP column (GE-Healthcare). Following extensive washing, PsiM was eluted using a 0–250 mM imidazole gradient in lysis buffer. Purification was verified by SDS-PAGE (Supplementary Fig. 6) and the protein quantified photometrically at 280 nm. For subsequent use in activity assays, the enzyme was subjected to buffer exchange on a PD-10 column (Cytiva) eluted with 50 mM Tris (pH 8.4) or the respective buffers used to determine the pH optimum (below).

### Production of PsiM N247M for binding studies and activity assays

Using pJF45 as the template, a gene encoding the PsiM N247M mutant was generated by overlap extension PCR[78] and inserted into pET-28a (Novagen) by means of standard cloning techniques. The resulting construct was verified by sequencing. Expression and purification followed the same protocol as described above for PsiM.

### Methyltransferase assays

Norbaeocystin, baeocystin and aeruginascin were obtained from the Usona Institute (Madison, WI, USA). Psilocybin was synthesised as described[13]. Methyltransferase activity assays were set up as triplicates in 50 mM Tris buffer (pH 8.4) in a total volume of 50 μl. The reactions were composed of 600 μM SAM and 200 μM acceptor substrate (norbaeocystin, baeocystin, or psilocybin). The reaction was started by adding PsiM (1 μM final concentration) and incubated at 28 °C (or between 14 and 50 °C to record the temperature optimum). Negative controls were run with heat-inactivated enzyme. The assays were stopped after 2 and 24 h (60 min for temperature optimum) in liquid nitrogen. To record pH optima, the reactions were set up in the following buffers (each 50 mM): MES (pH 6), phosphate (pH 7) Tris (pH 8–8.4), glycylglycine (pH 8.9), CAPSO (pH 9.5–11). To record kinetic

parameters, the reaction tubes were prewarmed to 28 °C before the prewarmed reaction mixture was added. The reactions for kinetics were set up in Tris buffer (pH 8.4) as well, but contained 100 μM – 1 mM acceptor substrate, 1 mM SAM, and 500 nM PsiM. Reactions were stopped in liquid nitrogen after 60, 90, 120, 180, 300, and 600 s. The samples were lyophilised, resuspended in 50 μl methanol and subsequently placed in an ultrasonic bath for 3 min, followed by centrifugation.

Chromatographic analysis was performed on an Infinity II UHPLC-MS instrument (Agilent, Waldbronn, Germany) interfaced with a 6130 single quadrupole mass detector and electrospray ionisation source and equipped with an Ascentis Express 90 Å F5 column (100 × 2.1 mm, 2.7 μm particle size, kept at a constant temperature of 50 °C). Solvent A was 0.1% formic acid in water, solvent B was methanol. For elution, the following gradient was applied at a flow of 0.8 ml/min (0–2.5 min) and 0.6 ml/min (2.5–3.5 min): hold at 2% B (0–2 min), linearly increase to 20% B over the next 0.5 min, hold at this percentage for another 1 min.

### Isothermal titration calorimetry
ITC was performed at 25 °C in Tris buffer (50 mM, pH 8.4) using a Malvern Microcal PEAQ-ITC microcalorimeter. Titrants and protein concentration were: (i) 750 μM (norbaeocystin, baeocystin, SAH) to 50 μM PsiM, or (ii) 750 μM (norbaeocystin, baeocystin) to 1 mM sinefungin + 50 μM PsiM, or (iii) 750 μM (norbaeocystin, baeocystin) to 2 mM SAH + 50 μM PsiM, or (iv) Tris 50 mM buffer pH 8.4 to 50 μM PsiM. Results were analysed using the MicroCal PEAQ-ITC Analysis Software, assuming a single-site binding model.

### Sequence analysis and phylogenetics
Database searches to identify PsiM and METTL16/RlmF homologues were performed using the BLAST[32] server at NCBI[33]. The proteomes of representative species in the non-redundant protein sequence (nr) database were searched for sequences similar to *P. cubensis* PsiM and *H. sapiens* METTL16, respectively. A multiple sequence alignment was then produced with the help of MAFFT[79,80]. Phylogenetic analysis was carried out using the maximum-likelihood (ML) approach and JTT matrix-based model[81] in MEGA11[82]. Initial trees for the heuristic search were obtained automatically by applying the Neighbour-Join and BioNJ algorithms to a matrix of pairwise distances estimated using the JTT model, and by subsequently selecting the topology with optimal log likelihood value. A discrete Gamma distribution was used to model evolutionary rate differences among sites, allowing for 5 categories. All positions with less than 80% site coverage were eliminated, *i.e.*, fewer than 20% alignment gaps were allowed at any position (partial deletion option). There were a total of 280 positions in the final dataset. Bootstrap analysis was performed with 1000 repeats.

### Reagents
A list of reagents and oligonucleotide primers is provided in Supplementary Table 2.

### Reporting summary
Further information on research design is available in the Nature Portfolio Reporting Summary linked to this article.

## Data availability
All crystallographic structures and corresponding diffraction data have been deposited in the PDB, with the following accession numbers: 8PB3 (PsiM in complex with SAH and norbaeocystin, monoclinic crystal form), 8PB4 (PsiM in complex with SAH and norbaeocystin, orthorhombic crystal form), 8PB5 (PsiM in complex with sinefungin and norbaeocystin), 8PB6 (PsiM in complex with SAH and baeocystin), 8PB7 (PsiM in complex with sinefungin and baeocystin), 8PB8 (PsiM in complex with SAH) and 8QXQ (PsiM in complex with SAH and psilocybin). The previously reported structures used in this work are also available in the PDB: 6B92 (Crystal Structure of the N-terminal domain of human METTL16 in complex with SAH), 6DU4 (Crystal structure of hMettl16 catalytic domain in complex with MAT2A 3'UTR hairpin 1), 6GFK (delta-N METTL16 MTase domain). The enzymatic activity data generated in this study are provided in the Source Data file. Source data are provided with this paper.

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

## Acknowledgements
We thank the beamline staff at the European Synchrotron Radiation Facility (ESRF) in Grenoble, France, for their kind assistance during X-ray data collection. We are grateful to Andrea Perner and Janis Fricke (Leibniz Institute for Natural Product Research and Infection Biology, Hans-Knöll-Institute, Jena, Germany) for recording HRMS spectra and construction of expression vectors, respectively. We also gratefully acknowledge Alexander Sherwood and Robert Kargbo (Usona Institute, Madison, WI, USA) for providing norbaeocystin, baeocystin and aeruginascin. This work was supported by the Deutsche Forschungsgemeinschaft (DFG grant HO2515/11-1 to D.H.) and by the Austrian Science Fund (FWF grant I-5192 to B.R., DOI 10.55776/I5192). J.H. was the recipient of a Fulbright-Austrian Marshall Plan Foundation Award.

## Author contributions
B.R. and D.H. initiated the project and obtained funding; S.W., J.H. and K.R. produced and purified proteins; S.W. obtained crystals; S.W. and J.H. collected X-ray diffraction data; S.W., J.H. and B.R. solved and refined structures; K.R. and M.M. characterised interactions by ITC; K.R., J.H. and S.D. performed enzymatic assays; S.W. analysed sequence data; S.W. and J.H. performed mutagenesis; S.W. coordinated the project and wrote the manuscript.

## Competing interests
The authors declare no competing interests.
