## [Peer Review File · Nature Communications]

Methyl Transfer in Psilocybin BiosynthesisREVIEWER COMMENTS

Reviewer #1 (Remarks to the Author):

Hudspeth et al. describe the crystal structure of the methyltransferase (MT) PsiM in complex with several ligand combinations (SAH, SAH and norbaeocystin or baeocystin, sinefugin and norbaeocystin). Based on these structures, the authors identify two substrate binding modes (one productive, the other likely unproductive), a plausible explanation as to why the enzyme catalyzes demethylation but not trimethylation, and provide a structural rationalization for the substrate promiscuity observed in an earlier study (Ref. 46). The crystal structure also confirms predictions based on primary sequence analysis that PsiM is directly related to nucleic acid MTs. Interestingly, the authors found that the phosphate-binding motif of PsiM is the same as the phosphate-binding-motif in nucleic acid MTs, providing an excellent illustration that phosphate-binding is a very strong evolutionary signature that is often conserved among proteins with similar fold but different catalytic activities.

PsiM is an important enzyme because of the growing interest in psilocybin as a therapeutic and the possibilities to produce psilocybin analogs by biocatalysis. The crystallographic data is of excellent quality and the described structures provide relevant insight into the substrate and product specificity of this enzyme.

On the other hand, the authors hardly fulfill their claim to provide new insight into the catalytic mechanism of this enzyme. This is indeed difficult for a predominantly crystallographic study but the authors should at least discuss their high-resolution structures in light of the recent mechanistic literature on N-methyltransferases including:

- 10.3390/molecules23112965
- 10.1021/acs.biochem.8b01141
- 10.1021/acscchembio.5b00852
- 10.1021/acs.biochem.1c00028

Since the authors avoid any discussion of the relevant literature it is quite bold to claim that their work provides “a paradigm for the function of related SAM-dependent methyltransferases”.

Also, for a high-impact publication one would expect that that authors demonstrate that their structure supports “bioengineering efforts” by generating variants that do accept alternative substrates.

Minor comments:

Abstract: “Inherent limitations of the ancestral monomethyltransferase scaffold hamper the efficiency of psilocybin assembly and leave PsiM incapable of catalysing trimethylation to aeruginascin, highlighting persistent consequences of evolutionary tinkering.” I find this argument throughout the manuscript rather weak. Although the structure of PsiM provides an explanation why a third methylation is

inefficient, it says absolutely nothing about evolutionary pressures. The authors are free to maintain their opinion, but it is not a strong position.

L56: “ultimate” ◇ last

L62: “In the present study, we provide a comprehensive structural, biochemical and phylogenetic characterisation of PsiM, shedding light on its reaction mechanism, specificity and unusual evolutionary history” The authors must specify as to what mechanistic aspect they believed to have shed light on.

L101: “These observations are corroborated...” ◇ rationalized or explained (an observation technically does not need corroboration, but explanation)

L112: “Thus, coenzyme binding, substrate recognition and catalysis are inextricably linked.” This statement is very generally correct. The authors might want to be more specific as to what should be highlighted as novel insight.

L113: “Consistent with earlier predictions¹⁸, the region following strand β 4, which lacks homology to other methyltransferases, acts as a substrate recognition loop (SRL). Comprising residues 189-221, the SRL forms a highly contorted structure, stabilised by its interaction with norbaeocystin and by intramolecular hydrogen bonds that facilitate turns.” This is hard to follow. Please rephrase.

L131: “isothermal....” Since the data is available the authors should discuss the different complex stabilities.

L154 – 164: “An alternative state...sinefungin...similarity between sinefungin and SAM... analogous dynamic equilibrium...in the PsiM-SAMnorbaeocystin” The real take home message is that sinefungin is a terrible substrate analog with regard to mechanistic studies.

L165: “Although we anticipated that this particular combination might enable the reverse reaction (regenerating norbaeocystin and SAM)” This is a very remote assumption.

L172: “hit and run” there is no need to invent new terms here. The result shows that crystallography is unable to “characterise the state of the active site directly after the first methyl transfer”. Apparently, methyl transfer results in a product:SAH complex that relaxes to the observed complex most likely due to electrostatic interactions.

L178: “engaged in a catalytic hydrogen bond” → delete “catalytic”

L179: “a configuration with the methyl” ◇ conformation

L179: “The only conceivable alternative, a configuration with the methyl group pointing towards the carbonyl of Pro185, is not observed at all, presumably because the conformationally constrained backbone of the NPPF motif is unable to move aside.” Introduction of mutations that lessen this conformational constraint could enable the enzyme to catalyze trimethylation, if an evolutionary pressure to produce the trimethylated product existed. This is why I don't buy the statement in L34

(abstract).

L181: "Intriguingly, we find that substrate occupancy is only 61% in this complex, with a large portion of the SRL (residues 198-208), Arg281, as well as numerous associated water molecules showing signs of disorder in the absence of baecocystin." Not clear, rephrase

L185: "presence of SAM analogues such as sinefungin" What other analogs were used?

L188: "large amounts" use specific number please

L189: "Quantitative experiments (Fig. 4b) are consistent with Michaelis-Menten kinetics and...." MM kinetic is supposed to be quantitative too

L190: "that the first methyl transfer is approximately twice as fast as the second, with k_{cat} values of 0.09 +/- 0.01 min⁻¹ and 0.05 +/- 0.01 min⁻¹." Comparing catalytic efficiencies would be more relevant here.

L193: "Taken together, these observations demonstrate that PsiM is a comparatively poor dimethyltransferase." $(0.09/340)/(0.05/360) = 1.9$. The catalytic efficiencies differ by no more than 2-fold! This is a small difference considering that MTs accelerate methyl transfer by 10000000000-fold over the uncatalyzed reaction. It is even possible that close PsiM homologs catalyze the second methylation more efficiently.

L196: "Dali27" move reference numbers to the end of a sentence.

L198: "Z-scores ranging from 25 to 35" Explain Z-scores

L204: "to our knowledge" replace this lazy comment. Name the data base that was searched without hits.

L213: "perhaps most intriguingly, only transfer a single methyl group to their target amine" Why is RNA N-methylated in the first place? Dimethylation may not be required or even harmful.

L214: "A likely explanation is provided" explanation for what?

L245: "In vitro assays did not appear to yield aeruginascin either¹⁰." Rephrase

L259: "The crystallographic structures of PsiM presented here reveal the detailed reaction mechanism of psilocybin biosynthesis and provide a paradigm for the function of related SAM-dependent methyltransferases." It is not clear what the authors mean with "detailed reaction mechanism". Please specify.

L261: "structures confirm the "proximity and desolvation" What do the authors mean with desolvation? The amino group of the substrate is perfectly well solvated by H-bonding.

L264: "In particular, we find alternative substrate conformations of likely functional significance in nearly

all of the structures.” Not clear what the authors mean with “likely functional significance”

L265: “Our study also sheds a new light on the evolution of SAM-dependent methyltransferases and particularly on their ability to adapt to novel substrates.” Are there other natural product MTs that emerged from a nucleic acid MT?

L274: “considerably less efficient than the first” incorrect

L275: “This suggests that the enzyme is caught in local optimum, an “evolutionary cul-de-sac”, as a consequence of the inherent limitations of its original scaffold.” Very difficult opinion to maintain

L277: “efficiently di- and trimethylate amines, such as the histidine methyltransferase EgtD37,38” Discuss other examples with different evolutionary origins.

L281: “It thus seems plausible that the rigid geometry of the proline-rich sequence plays a role in imposing sp³ hybridisation, rather than sp², while guiding the lone pair towards the methyl group of SAM.” This link between sequence motifs and hybridization of particular substrate atoms is weak at best.

L82: “the amines in the psilocybin precursors norbaeocystin and baeocystin are not planar” What do you mean with a planar amine?

L285: “The apparent pressure on hallucinogenic mushrooms to evolve and maintain a pathway culminating in the tertiary amine psilocybin, i.e., the dimethylated species, is rooted in the properties of psilocin and the self-protection mechanism of the fungus against this dephosphorylated shunt product” makes no sense. “shunt product” is this really the right term here?

LL293: “pseudo-ring formation” given the myriads of intramolecular H-bonds in natural and synthetic compounds one should be alarmed by the rare usage of the term “pseudo-ring formation” and the type of publications it appears. This is an unfortunate term.

L302: “Given that PsiM and the methyltransferase domain of human METTL16 are highly similar – their conserved domains retain 37% sequence identity – our results provide direct insight into the catalytic mechanism of the latter.” Rephrase. “highly similar” with regards to what? Structure or sequence?

L306: “presumably owing to a physiologically relevant auto-inhibitory mechanism” not clear.

LL308: “Our structures of PsiM can be used as templates to accurately model ternary complexes of METTL16.” Accurately – how would you know?

L317: “Perhaps most intriguingly, METTL16 was shown to bind RNA before it recruits SAM45, whereas our results clearly indicate that PsiM must recruit the cofactor before it is able to interact with its substrates.” Studying tThe literature will tell the authors that this may be a pseudo-difference. Early recognition between nucleic acid-MTs and nucleic acid may occur via residues outside the active site. Specific binding of the targeted nucleobase may still occur after binding of SAM. The microscopic

mechanism of these enzymes is likely similar to that of PsiM.

L326: "These structure-based predictions are corroborated by recent experiments involving alternative substrates in psilocybin-producing *E. coli* strains⁴⁶." A prediction after the fact is no longer a prediction. The current results provide explanations for the earlier findings.

L328: "As the relevance of psilocybin as a future medication against a growing number of common mental health conditions is becoming clear," add a reference

Reviewer #2 (Remarks to the Author):

In their article, "Methyl Transfer in Psilocybin Biosynthesis", Hudspeth and colleagues report high resolution diffraction structures of the enzyme, PsiM, which is responsible for the dimethylation of norbaeocystin in the psilocybin biosynthetic pathway. The authors analyze PsiM structures at several stages of its reaction cycle, represented through co-crystallization with different substrates and cofactors. This structural work is followed by a phylogenetic analysis in which the authors provide evidence for the evolution of PsiM dimethylation from the family of METTL16 RNA methyltransferases.

Overall, this is a fascinating manuscript which provides both an important advance in the enzymology of PsiM and an interesting case study for mechanisms of protein evolution more broadly. The manuscript is very well written and presents the experiments and their results clearly. I cannot evaluate the crystallographic work as this is outside of my area of expertise, but I do have several comments regarding the evolutionary analysis that I suggest the reviewers address in a revised manuscript.

1) Presumably, *Psilocybe* sp have also retained their METTL16 enzymes, but these are not present on the tree in Fig 6. A thorough evolutionary analysis would include these and all PsiM/METTL16/RlmF homologs from the species represented on the tree. By comparing the resulting tree to a high quality species tree, the authors should, in theory, be able to identify whether PsiM resulted from neofunctionalization following a protein duplication and where that protein duplication took place on the species tree. I suggest a that the phylogenetic analysis be performed again following a broader sequence collection strategy (at least within the *Psilocybe* sp and other fungi represented on the tree). While doing so, the authors should also take care to include only true PsiM/METTL16 homologs and not other proteins with a shared domain, e.g. a Rossmann fold, that are otherwise not homologous.

2) The description of the sequence analysis and phylogenetics methods is insufficient in the current version of the manuscript. The description of the sequence search strategy should include the algorithm that was used, the query sequence(s) that were used, the sequence database that was searched, and whether certain taxonomic groups were explicitly included or excluded from the search. The structural comparison method should also be included and should mention the number of C-alpha atoms that were used in the RMSD calculation between PsiM and METTL16 as RMSD values can be sensitive to the

number of atoms being compared in the superposition.

3) The sequence alignment editing strategy seems extreme to me, i.e., the removal of every sequence positions containing a gap. What percentage of the total alignment does this edit represent and did it remove any of the domains and motifs discussed in previous sections of the manuscript? It would be useful to include the original sequence alignment with the edited areas highlighted (either in the main document or the supplement).

4) Small typo. Line 323: "likely to be be tolerated"

Reviewer #3 (Remarks to the Author):

Psilocybin, the natural hallucinogen, is an indole alkaloid produced by magic mushrooms and has the potential to treat psychological conditions. The final step of psilocybin synthesis is catalyzed by PsiM, which dimethylates the primary amine of the precursor norbaeocystine via the monomethylated intermediate baeocystin to produce the ternary amine psilocybin.

In this manuscript, Hudspeth et al. describe the detailed molecular mechanism of dimethylation of norbaeocystine through extensive structural analysis of PsiM in complex with various substrates (psilocybin, norbaeocystin, baeocystin, SAH, sinefungin) at high resolutions. The structural and biochemical analysis allowed the authors to propose the detailed sequential reaction mechanism of dimethylation of norbaeocystin via baeocystin to produce psilocybin by PsiM. The structures also provide why PsiM does not trimethylate norbaeocystine to produce aeruginascin. The structure of PsiM is homologous to that of METTL16, which methylates specific adenine residues in specific RNAs (U6 snRNA and Mat2A mRNA), and phylogenetic analysis supports the idea that PsiM evolved from METTL16. Comparison of the structures of PsiM complexed with norbaeocystin and METTL16 complexed with RNA suggests that the PsiM substrate, norbaeocystin, mimics RNA. Furthermore, based on the structural comparison, a single amino acid mutation in the catalytic pocket converts PsiM to catalyze monomethylation, but not demethylation of norbaeocystin.

The structural and biochemical analysis in this manuscript is solid. The structural information in this manuscript provides important information to develop PsiM with improved therapeutic properties. This review has several comments that may improve the manuscript.

Comments.

1) Page 10, Line 73:

"Although a single substitution, M247N, enables PsiM to catalyze dimethylation, the second methyltransfer remains considerably less efficient than the first." Is this sentence correct? The sentence should be, "Although the presence of N247 in PsiM, instead of M274 as in METTL16, enables PsiM to catalyze dimethylation..."

2) Page 11, Line 287:

Is the wording "shunt product" common?

3) Page 12, Line 313:

The activity of METTL16 is tightly regulated by the SAM concentration. In human METTL16, the "K-loop" is proposed to be involved in this regulation (Ref. 28). It would be helpful to display the amino acid sequence alignments in the K-loop and discuss them in the text. Are the sequences corresponding to the K-loop conserved?

4) Related to 3) above. The K_m value for SAM in the methylation of RNA by human METTL16 (and warm homolog METT19) is relatively high (~several hundreds of micromolar). If the K_m value for SAM in the methylation of norbaeocystin by PsiM is available, the authors could discuss the conservation of the K-loops between METTL16 and PsiM.

5) Conservation of the SRLs between METTL16 and PsiM could be mentioned or discussed in the text. As mentioned above, the sequence alignments would be useful.

6) The N-terminal Jaw domains in human METTL16 and warm METT10 are required for RNA binding (Ref. 28 and NAR 51 2434-2446). Does the deletion of the Jaw domain from PsiM affect its methylation activity?

Reviewer #1

[...], the authors hardly fulfill their claim to provide new insight into the catalytic mechanism of this enzyme. This is indeed difficult for a predominantly crystallographic study but the authors should at least discuss their high-resolution structures in light of the recent mechanistic literature on N-methyltransferases including:

- 10.3390/molecules23112965
- 10.1021/acs.biochem.8b01141
- 10.1021/acscchembio.5b00852
- 10.1021/acs.biochem.1c00028

Since the authors avoid any discussion of the relevant literature it is quite bold to claim that their work provides “a paradigm for the function of related SAM-dependent methyltransferases”.

We have now included a discussion of our structures in the light of the recent studies suggested by reviewer 1. In addition, the wording that the reviewer felt was somewhat bold (“paradigm”, etc.) has been toned down. We ought to point out, however, that the resolution attained in our crystallographic study is (by a fair margin) the highest for any methyltransferase currently represented in the PDB. As such, the structures are key reference points for future work on such enzymes.

Minor comments:

Abstract: “Inherent limitations of the ancestral monomethyltransferase scaffold hamper the efficiency of psilocybin assembly and leave PsiM incapable of catalysing trimethylation to aeruginascin, highlighting persistent consequences of evolutionary tinkering.” I find this argument throughout the manuscript rather weak. Although the structure of PsiM provides an explanation why a third methylation is inefficient, it says absolutely nothing about evolutionary pressures. The authors are free to maintain their opinion, but it is not a strong position.

It is in fact the second methylation reaction that is less efficient, while the third does not take place at all; the structures indeed explain these observations. Our hypothesis that the functional limitations of PsiM are a consequence of its evolutionary origins does not stem from an assessment of selection pressures. It is suggested by the fact that none of PsiM’s immediate ancestors are able to di- or trimethylate, plus the observation that sequence and structure have been highly conserved in PsiM. Moreover, the catalytic properties of PsiM contrast with those of typical small-molecule N-di/trimethyltransferases, which tend to have different active site architectures and, unlike PsiM, are able to exchange SAH/SAM for the second/third round of methylation while the intermediate product remains bound to the active site. We have further clarified these arguments in the manuscript.

L56: “ultimate” → last

Text changed.

L62: “In the present study, we provide a comprehensive structural, biochemical and phylogenetic characterisation of PsiM, shedding light on its reaction mechanism, specificity and unusual evolutionary history” The authors must specify as to what mechanistic aspect they believed to have shed light on.

As this line is part of the introduction section, we do not think we should provide a more detailed account of the results here. However, a paragraph has been added to the discussion section to emphasise the mechanistic aspects of our study.

L101: “These observations are corroborated...” → rationalized or explained (an observation technically does not need corroboration, but explanation)

Text changed accordingly.

L112. “Thus, coenzyme binding, substrate recognition and catalysis are inextricably linked.” This statement is very generally correct. The authors might want to be more specific as to what should be highlighted as novel insight.

Text changed.

L113: “Consistent with earlier predictions¹⁸, the region following strand β 4, which lacks homology to other methyltransferases, acts as a substrate recognition loop (SRL). Comprising residues 189-221, the SRL forms a highly contorted structure, stabilised by its interaction with norbaeocystin and by intramolecular hydrogen bonds that facilitate turns.” This is hard to follow. Please rephrase.

Text has been rephrased.

L131: “isothermal...” Since the data is available the authors should discuss the different complex stabilities.

The experimentally determined complex stabilities are now compared in the text.

L154 – 164: “An alternative state...sinefungin...similarity between sinefungin and SAM... analogous dynamic equilibrium...in the PsiM-SAMnorbaeocystin” The real take home message is that sinefungin is a terrible substrate analog with regard to mechanistic studies.

Sinefungin is chemically distinct from SAM – inevitably, as it is an inhibitor. Sterically, however, sinefungin mimics SAM in the sense that, unlike SAH, it fills the space otherwise occupied by the transferable methyl group. In the case of the baeocystin complex, this imposes the same substrate conformation as in the SAM complex, revealing that the first methyl turns towards Asn183 and not towards Pro185. We do agree that the result with norbaeocystin is less easy to interpret. Instead of a hydrogen bond between the norbaeocystin and sinefungin amines (which we observe and discuss), the pre-transition state with SAM might be expected to feature a tetrel bond between the norbaeocystin amine and the transferable methyl group. Although a typical tetrel bond is shorter than the hydrogen bond that we observe, it is normally longer than the 1.7–1.8 Å separating the amine from the predicted methyl carbon position in our structures. One way of initially accommodating the longer distance, in a hypothetical pre-transition state, would be by means of alternative conformations analogous to the ones we observe in the presence of sinefungin. We agree that this model is speculative, and we have adapted the text to reflect this.

L165: “Although we anticipated that this particular combination might enable the reverse reaction (regenerating norbaeocystin and SAM)” This is a very remote assumption.

This line of thought was not essential and has been removed.

L172: “hit and run” there is no need to invent new terms here. The result shows that crystallography is unable to “characterise the state of the active site directly after the first methyl transfer”. Apparently, methyl transfer results in a product:SAH complex that relaxes to the observed complex most likely due to electrostatic interactions.

The term “hit and run” has been removed.

L178: “engaged in a catalytic hydrogen bond” → delete “catalytic”

Text changed accordingly.

L179: “a configuration with the methyl” → conformation

Text changed accordingly.

L179: “The only conceivable alternative, a configuration with the methyl group pointing towards the carbonyl of Pro185, is not observed at all, presumably because the conformationally constrained backbone of the NPPF motif is unable to move aside.” Introduction of mutations that lessen this conformational constraint could enable the enzyme to catalyze trimethylation, if an evolutionary pressure to produce the trimethylated product existed. This is why I don't buy the statement in L34 (abstract).

The reviewer seems to assume here that the NPPF motif is an independent unit, elements of which can be freely mutated (and rendered more flexible) independently of the over-all architecture of the active site. This is clearly not the case, as any mutation of the central proline residues is known to completely inactivate NPPF-containing enzymes. In other words, the inherent rigidity of the NPPF-motif appears to be essential to the particular type of enzyme scaffold associated with it.

L181: “Intriguingly, we find that substrate occupancy is only 61% in this complex, with a large portion of the SRL (residues 198-208), Arg281, as well as numerous associated water molecules showing signs of disorder in the absence of baecocystin.” Not clear, rephrase

Text was rephrased.

L185: “presence of SAM analogues such as sinefungin” What other analogs were used?

None, text has been changed.

L188: “large amounts” use specific number please

Exact numbers are not available, but the text was rephrased so as to avoid the uninformative expression “large amounts”.

L189: “Quantitative experiments (Fig. 4b) are consistent with Michaelis-Menten kinetics and...”
MM kinetic is supposed to be quantitative too

No criticism of Michaelis-Menten theory intended, text rephrased.

L190: “that the first methyl transfer is approximately twice as fast as the second, with k_{cat} values of $0.09 \pm 0.01 \text{ min}^{-1}$ and $0.05 \pm 0.01 \text{ min}^{-1}$.” Comparing catalytic efficiencies would be more relevant here.

Catalytic efficiencies have been added to the text.

L193: “Taken together, these observations demonstrate that PsiM is a comparatively poor dimethyltransferase.” $(0.09/340)/(0.05/360) = 1.9$. The catalytic efficiencies differ by no more than 2-fold! This is a small difference considering that MTs accelerate methyl transfer by 10000000000-fold over the uncatalyzed reaction. It is even possible that close PsiM homologs catalyze the second methylation more efficiently.

*The 2-fold difference in itself may not seem huge, but the observation does contrast with data for typical small-molecule di- and trimethyltransferases (several examples of which are now provided in the text) that **preferentially** catalyse the second/third reaction. Unlike such enzymes, PsiM also needs to let go of the monomethylated intermediate before it can release SAH and bind SAM for the subsequent round of catalysis. The combined effect is that considerable amounts of the monomethylated species (baeocystin) accumulate.*

L196: “Dali27” move reference numbers to the end of a sentence.

Clarity was improved by rephrasing the sentence and moving the reference number.

L198: “Z-scores ranging from 25 to 35” Explain Z-scores

The Z-score in question is an indicator of structural similarity generated by the program Dali. Values greater than ~2 are generally considered significant. A detailed explanation of the metric can be found in the literature describing Dali, which we refer to in the manuscript.

L204: “to our knowledge” replace this lazy comment. Name the data base that was searched without hits.

The data base that was searched has been added, together with the appropriate references.

L213: “perhaps most intriguingly, only transfer a single methyl group to their target amine” Why is RNA N-methylated in the first place? Dimethylation may not be required or even harmful.

The biological function of RNA N-methylation (regulation of gene expression and splicing) is mentioned in the manuscript. What is intriguing here is not so much the observation that RNA is merely monomethylated by METTL16, but rather the fact that PsiM somehow acquired the ability to dimethylate, even though its sequence and structure are highly conserved with respect to METTL16. The text has been adapted to make this clearer.

L214: “A likely explanation is provided” explanation for what?

An explanation for the surprising evolutionary relationship between PsiM and METTL16. The text was rephrased to make this clearer.

L245: “In vitro assays did not appear to yield aeruginascin either10.” Rephrase

Rephrased.

L259: “The crystallographic structures of PsiM presented here reveal the detailed reaction mechanism of psilocybin biosynthesis and provide a paradigm for the function of related SAM-dependent methyltransferases.” It is not clear what the authors mean with “detailed reaction mechanism”. Please specify.

A more detailed discussion of findings relevant to the reaction mechanism has been added.

L261: “structures confirm the “proximity and desolvation” What do the authors mean with desolvation? The amino group of the substrate is perfectly well solvated by H-bonding.

The term “proximity and desolvation mechanism” is used in the literature that we cite. A concise explanation has been added to the text.

L264: “In particular, we find alternative substrate conformations of likely functional significance in nearly all of the structures.” Not clear what the authors mean with “likely functional significance”

The potential functional significance of various alternative states observed in our crystal structures is now discussed in greater detail.

L265:” Our study also sheds a new light on the evolution of SAM-dependent methyltransferases and particularly on their ability to adapt to novel substrates.” Are there other natural product MTs that emerged from a nucleic acid MT?

Not that we know of.

L274: “considerably less efficient than the first” incorrect

Removed “considerably”.

L275: “This suggests that the enzyme is caught in local optimum, an “evolutionary cul-de-sac”, as a consequence of the inherent limitations of its original scaffold.” Very difficult opinion to maintain

We have further clarified our reasoning in the text.

L277: “efficiently di- and trimethylate amines, such as the histidine methyltransferase EgtD37,38” Discuss other examples with different evolutionary origins.

Several other examples with different evolutionary origins have been included.

L281: “It thus seems plausible that the rigid geometry of the proline-rich sequence plays a role in imposing sp³ hybridisation, rather than sp², while guiding the lone pair towards the methyl group of SAM.” This link between sequence motifs and hybridization of particular substrate atoms is weak at best.

The link between the NPPF sequence motif and sp² hybridisation of substrate amines was described by Schubert et al. (Trends Biochem. Sci. 28, 329-335). The rule holds rather well: with the notable exception of PsiM, the NPPF motif has only been found in methyltransferases acting on this specific type of amine.

L82: “the amines in the psilocybin precursors norbaeocystin and baecocystin are not planar” What do you mean with a planar amine?

Not part of a conjugated system, text changed.

L285: “The apparent pressure on hallucinogenic mushrooms to evolve and maintain a pathway culminating in the tertiary amine psilocybin, i.e., the dimethylated species, is rooted in the properties of psilocin and the self-protection mechanism of the fungus against this dephosphorylated shunt product” makes no sense. “shunt product” is this really the right term here?

This phrase has been removed and the unusual term “shunt product” is no longer used anywhere in the manuscript.

LL293: “pseudo-ring formation” given the myriads of intramolecular H-bonds in natural and synthetic compounds one should be alarmed by the rare usage of the term “pseudo-ring formation” and the type of publications it appears. This is an unfortunate term.

The designation “pseudo-ring formation” was coined in an earlier publication by others, but we agree with the reviewer. The term has been removed.

L302: “Given that PsiM and the methyltransferase domain of human METTL16 are highly similar – their conserved domains retain 37% sequence identity – our results provide direct insight into the catalytic mechanism of the latter.” Rephrase. “highly similar” with regards to what? Structure or sequence?

Highly similar in sequence as well as in structure. Sentence rephrased.

L306: “presumably owing to a physiologically relevant auto-inhibitory mechanism” not clear.

Statement removed.

LL308: “Our structures of PsiM can be used as templates to accurately model ternary complexes of METTL16.” Accurately – how would you know?

The superfluous adverb has been removed.

L317: “Perhaps most intriguingly, METTL16 was shown to bind RNA before it recruits SAM45, whereas our results clearly indicate that PsiM must recruit the cofactor before it is able to interact with its substrates.” Studying tThe literature will tell the authors that this may be a pseudo-difference. Early recognition between nucleic acid-MTs and nucleic acid may occur via residues outside the active site. Specific binding of the targeted nucleobase may still occur after binding of SAM. The microscopic mechanism of these enzymes is likely similar to that of PsiM.

We have added this interesting possibility to the discussion.

L326: “These structure-based predictions are corroborated by recent experiments involving alternative substrates in psilocybin-producing E. coli strains⁴⁶.” A prediction after the fact is no longer a prediction. The current results provide explanations for the earlier findings.

Text changed accordingly.

L328: “As the relevance of psilocybin as a future medication against a growing number of common mental health conditions is becoming clear, ” add a reference

References to several recent publications have been added.

Reviewer #2

Overall, this is a fascinating manuscript which provides both an important advance in the enzymology of PsiM and an interesting case study for mechanisms of protein evolution more broadly. The manuscript is very well written and presents the experiments and their results clearly.

We would like to thank reviewer 2 for these remarks.

1) Presumably, Psilocybe sp have also retained their METTL16 enzymes, but these are not present on the tree in Fig 6. A thorough evolutionary analysis would include these and all PsiM/METTL16/RlmF homologs from the species represented on the tree. By comparing the resulting tree to a high quality species tree, the authors should, in theory, be able to identify whether

PsiM resulted from neofunctionalization following a protein duplication and where that protein duplication took place on the species tree. I suggest that the phylogenetic analysis be performed again following a broader sequence collection strategy (at least within the *Psilocybe* sp and other fungi represented on the tree). While doing so, the authors should also take care to include only true PsiM/METTL16 homologs and not other proteins with a shared domain, e.g. a Rossmann fold, that are otherwise not homologous.

This is an interesting idea and we have looked into it. We found that METTL16 itself has indeed been conserved in PsiM-producing species, and the sequences in question have now been included in the phylogenetic analysis. It is clear from the results that PsiM and METTL16 (i.e. from the species that have both) belong to distinct branches, suggesting that PsiM indeed arose by gene duplication prior to the diversification of the PsiM-producing species. On the other hand, pinpointing the event in an evolutionary tree of fungi is not as simple as it might seem. PsiM/METTL16 sequences alone do not yield a reliable and sufficiently detailed evolutionary tree of mushroom taxa and, consequently, it is impossible to determine with any confidence where exactly PsiM branched off from METTL16 proper. Adding large numbers (~100) of METTL16 sequences from a representative and highly diverse subset of mushrooms did not solve this problem.

2) The description of the sequence analysis and phylogenetics methods is insufficient in the current version of the manuscript. The description of the sequence search strategy should include the algorithm that was used, the query sequence(s) that were used, the sequence database that was searched, and whether certain taxonomic groups were explicitly included or excluded from the search. The structural comparison method should also be included and should mention the number of C-alpha atoms that were used in the RMSD calculation between PsiM and METTL16 as RMSD values can be sensitive to the number of atoms being compared in the superposition.

These methods are now described in greater detail. The number of superimposed C-alpha atoms has also been added to the text.

3) The sequence alignment editing strategy seems extreme to me, i.e., the removal of every sequence positions containing a gap. What percentage of the total alignment does this edit represent and did it remove any of the domains and motifs discussed in previous sections of the manuscript? It would be useful to include the original sequence alignment with the edited areas highlighted (either in the main document or the supplement).

For the revised manuscript, we have carried out the analysis with a mere partial removal of the gap regions (80% cut-off). There is no difference in the resulting tree topology. The complete alignment has been added to the supplementary material.

4) Small typo. Line 323: "likely to be be tolerated"

Corrected.

Reviewer #3

The structural and biochemical analysis in this manuscript is solid. The structural information in this manuscript provides important information to develop PsiM with improved therapeutic properties.

We would like to thank reviewer 3 for these comments.

1) Page 10, Line 73:

"Although a single substitution, M247N, enables PsiM to catalyze dimethylation, the second methyltransfer remains considerably less efficient than the first." Is this sentence correct? The sentence should be, "Although the presence of N247 in PsiM, instead of M274 as in METTL16, enables PsiM to catalyze dimethylation..."

The text has been changed accordingly.

2) Page 11, Line 287:

Is the wording "shunt product" common?

Removed.

3) Page 12, Line 313:

The activity of METTL16 is tightly regulated by the SAM concentration. In human METTL16, the "K-loop" is proposed to be involved in this regulation (Ref. 28). It would be helpful to display the amino acid sequence alignments in the K-loop and discuss them in the text. Are the sequences corresponding to the K-loop conserved?

A complete alignment of the MT domains of PsiM, METTL16 and RlmF proteins has been added as supplementary figure 4, with the position of the K-loop and various other features indicated above the alignment. The K-loop of METTL16 and the functionally important residues Lys163 and Met167 in particular have not been conserved in PsiM. This suggests that the regulatory mechanism proposed for METTL16 does not exist in PsiM.

4) Related to 3) above. The K_m value for SAM in the methylation of RNA by human METTL16 (and warm homolog METT19) is relatively high (~several hundreds of micromolar). If the K_m value for SAM in the methylation of norbaeocystin by PsiM is available, the authors could discuss the conservation of the K-loops between METTL16 and PsiM.

We have not measured K_m values for SAM but binding constants (K_d) are available for SAH binding by PsiM ($66 \pm 49 \mu\text{M}$, our study) and SAM-binding by a METTL16-RNA complex ($126 \pm 6 \mu\text{M}$, Breger et al. 2022). It would therefore appear that both enzymes have a rather low affinity for the coenzyme, which is apparently not (or at least not exclusively) dependent on the K-loop.

5) Conservation of the SRLs between METTL16 and PsiM could be mentioned or discussed in the text. As mentioned above, the sequence alignments would be useful.

An annotated sequence alignment has been added to the supplementary material. The SRL is not conserved between PsiM and METTL16, as would be expected given that the structure and function of this region have completely changed in the course of evolution. In METTL16 the SRL recognises a stem-loop structure in the target RNA, whereas in PsiM it forms a lid that closes off the binding pocket and tethers the small-molecule substrate. This striking difference is illustrated in Fig. 6A.

6) The N-terminal Jaw domains in human METTL16 and warm METT10 are required for RNA binding (Ref. 28 and NAR 51 2434-2446). Does the deletion of the Jaw domain from PsiM affect its methylation activity?

We have not attempted such a deletion experiment, as we expect the results to be difficult to interpret. The "jaw" domain is rather large (~70 residues) and interacts intimately with the core MT domain. As a result, the deletion in question would expose a sizeable hydrophobic surface, which is likely to affect the folding, stability and solubility of the resulting mutant.

REVIEWERS' COMMENTS

Reviewer #1 (Remarks to the Author):

In my view, the manuscript is now fit for publication.

Reviewer #2 (Remarks to the Author):

The authors have thoroughly revised this manuscript and all of my suggestions have been addressed.

Reviewer #3 (Remarks to the Author):

I am satisfied with the revision, except one response to the previous comment (6). The authors should provide the additional experiment using the protein lacking the jaw domain. This region is important for RNA recognition by METTL16 in human and METT10 in *C. elegans*.

In the abstract, the authors claimed that "Structural and phylogenetic analyses indicate that PsiM derives from epitranscriptomic N6-methyladenosine writers of the METTL16 family, which is explained by the observation that bound substrates physicochemically mimic RNA". Thus, the authors should examine this experiment. The response to the comment (6) does not justify for not performing the easy and quick experiment.

Reviewer #1

In my view, the manuscript is now fit for publication.

We thank the reviewer.

Reviewer #2

The authors have thoroughly revised this manuscript and all of my suggestions have been addressed.

We thank the reviewer.

Reviewer #3

I am satisfied with the revision, except one response to the previous comment (6). The authors should provide the additional experiment using the protein lacking the jaw domain. This region is important for RNA recognition by METTL16 in human and METT10 in *C. Elegans*. In the abstract, the authors claimed that “Structural and phylogenetic analyses indicate that PsiM derives from epitranscriptomic N6-methyladenosine writers of the METTL16 family, which is explained by the observation that bound substrates physicochemically mimic RNA”. Thus, the authors should examine this experiment. The response to the comment (6) does not justify for not performing the easy and quick experiment.

Rather than performing the experiment proposed by reviewer #3 (the outcome of which would most likely be inconclusive due to the effect the jaw domain is certain to have on overall enzyme stability), we have followed the suggestions from the editor and toned down our claims in the abstract.